# Impaired histone inheritance promotes tumor progression

Congcong Tian ®[1,13], Jiaqi Zhou ®[1,13], Xinran Li ®[1,13], Yuan Gao ®[2], Qing Wen ®[1], Xing Kang[1], Nan Wang[1], Yuan Yao[1], Jiuhang Jiang[1,3], Guibing Song[1,4], Tianjun Zhang[1,5], Suili Hu[1,3], JingYi Liao[1], Chuanhe Yu[6], Zhiquan Wang ®[7], Xiangyu Liu[8], Xinhai Pei ®[9], Kuiming Chan ®[10,11], Zichuan Liu[12] & Haiyun Gan ®[1] ✉

Faithful inheritance of parental histones is essential to maintain epigenetic information and cellular identity during cell division. Parental histones are evenly deposited onto the replicating DNA of sister chromatids in a process dependent on the MCM2 subunit of DNA helicase. However, the impact of aberrant parental histone partition on human disease such as cancer is largely unknown. In this study, we construct a model of impaired histone inheritance by introducing MCM2-2A mutation (defective in parental histone binding) in MCF-7 breast cancer cells. The resulting impaired histone inheritance reprograms the histone modification landscapes of progeny cells, especially the repressive histone mark H3K27me3. Lower H3K27me3 levels derepress the expression of genes associated with development, cell proliferation, and epithelial to mesenchymal transition. These epigenetic changes confer fitness advantages to some newly emerged subclones and consequently promote tumor growth and metastasis after orthotopic implantation. In summary, our results indicate that impaired inheritance of parental histones can drive tumor progression.

Chromatin states and their associated epigenetic information need to be faithfully inherited through cellular divisions to maintain cell identity[1,2]. Epigenetic aberrations are associated with a wide range of diseases, including cancer[3]. As important determinants for cellular epigenetic state, histone post-translational modifications (PTMs) carry epigenetic information and regulate gene transcription.

During chromatin replication, the evicted parental histones ahead of the replisome will be recycled to the newly replicated DNA by MCM2-POLA1 axis[4–7] and POLE3/4[7,8], along with newly synthesized histones be deposited by ASF1 and CAF-1[9–11]. Recent studies have reported that abnormal expression of CAF-1 can promote tumorigenesis and drive tumor metastasis[12–14]. However, even though several

[1]CAS Key Laboratory of Quantitative Engineering Biology, Guangdong Provincial Key Laboratory of Synthetic Genomics and Shenzhen Key Laboratory of Synthetic Genomics, Shenzhen Institute of Synthetic Biology, Shenzhen Institute of Advanced Technology, Chinese Academy of Sciences, 518055 Shenzhen, China. [2]Cold Spring Harbor Laboratory, Cold Spring Harbor, NY 11724, USA. [3]College of Veterinary Medicine, South China Agricultural University, 483 Wushan Road, 510642 Guangzhou, Guangdong, China. [4]College of Animal Science and Technology, Northwest A&F University, 712100 Shaanxi, Angling, China. [5]Department of Molecular and Biomedical Science, School of Biological Sciences, The University of Adelaide, Adelaide, SA 5005, Australia. [6]Hormel Institute, University of Minnesota, Austin, MN 55912, USA. [7]Division of Hematology, Department of Medicine, Mayo Clinic, Rochester, MN 55905, USA. [8]Guangdong Provincial Key Laboratory of Regional Immunity and Diseases, International Cancer Center, Marshall Laboratory of Biomedical Engineering, Shenzhen University Health Science Center, 518060 Shenzhen, China. [9]Department of Anatomy and Histology, Shenzhen University Health Science Center, 518060 Shenzhen, China. [10]Department of Biomedical Sciences, City University of Hong Kong, Hong Kong Special Administration Region, China. [11]Key Laboratory of Biochip Technology, Biotech and Health Centre, Shenzhen Research Institute of City University of Hong Kong, 518172 Shenzhen, China. [12]School of Pharmaceutical Science and Technology, Tianjin University and Health-Biotech United Group Joint Laboratory of Innovative Drug Development and Translational Medicine, Tianjin University, 300072 Tianjin, China. [13]These authors contributed equally: Congcong Tian, Jiaqi Zhou, Xinran Li. ✉e-mail: hy.gan@siat.ac.cn

genetic mutations in parental histone chaperones, such as MCM2[15,16] and POLA1[17] have been reported in a number of cancer patients, and POLE4 deficiency mice exhibit tumor predisposition[18], whether parental histone inheritance plays a role in tumorigenesis or tumor evolution is still unclear.

During DNA replication, the recycling of parental histone H3-H4 tetramers that carry the epigenetic information and do not split, is the most important step for chromatin state maintenance[19,20]. In mouse embryonic stem cells (mESCs), parental histones are partitioned almost equally onto both daughter DNA strands, with a weak leading-strand bias[6,7]. Parental histones carrying repressive marks (such as H3K27me3) are recycled to their original genomic locations by the replisome and act as seeds to re-establish histone PTMs in daughter cells via the postulated "read and write" mechanism[1,21–23]. The parental histones with active marks (such as H3K4me3) are also re-incorporated in close proximity to their original positions[19], which may enable transcription to be resumed quickly and accurately after DNA replication[24]. In turn, resumed transcription can drive transcription-coupled modification of new histones[25]. Thus, the active chromatin state is most likely maintained through mitosis, via both replication- and transcription-dependent histone deposition processes. However, whether a similar mechanism also applies to human cancer cells, and its role in tumor evolution is still poorly understood. Therefore, a tumor cell model featuring aberrant parental histone partition could enable us to investigate the role of parental histone inheritance in cancer biology.

In this study, we constructed a tumor model of impaired inheritance of parental histones by introducing an MCM2 histone-binding domain (HBD) mutation in the breast cancer cell lines. In this model, impaired histone inheritance results in dramatic epigenetic reprogramming, especially the pattern of the repressive histone mark H3K27me3, and promotes tumor growth and metastasis in vivo.

## Results

### Impaired histone inheritance leads to epigenetic reprogramming in breast cancer cell line MCF-7

It has been shown that mutating the HBD of MCM2 by substituting tyrosine (Y) 81 and 90 with alanine (A) impairs the transfer of parental histones to lagging strands without disturbing the helicase function of MCM2-7 complex in both *Saccharomyces cerevisiae* and mESCs[5,6,26]. However, the MCM2-dependent parental histone segregation pattern in cancer cells, as well as its role during tumor progression has not been investigated yet. Hence, we constructed an MCM2-2A mutant MCF-7 cell line without disturbing the expression and DNA replication function of MCM2 (Supplementary Fig. 1a–c) and characterized its pattern of parental histone segregation using enrichment and sequencing of protein-associated nascent DNA (eSPAN)[27]. H3K36me3 was commonly used in eSPAN experiments to monitor the parental histones H3 at leading and lagging strands of DNA replication forks[7,28]. In this study, we analyzed the bias pattern of H3K36me3 eSPAN around 7,624 human core replication origins in MCM2-2A mutant and wild-type (WT) MCF-7 cells (Fig. 1a, b and Supplementary Fig. 1d). Signals of H3K36me3 eSPAN exhibited a slight leading-strand bias in WT cells, whereas an exacerbated leading-strand bias in MCM2-2A mutant MCF-7 cells (Fig. 1b), which is consistent with previous observations in mESCs[7]. Moreover, we observed a similar exacerbated leading-strand bias in both MCM2-2A mutant HEK293T and MCM2-90A (Y90A) T47D cell lines (Supplementary Fig. 1e–h). These results indicate that during DNA replication, parental histones are normally assembled onto both leading and lagging strands, with a slight preference for leading strands in WT breast cancer and noncancerous differentiated cell lines. The transfer of parental histones to the lagging strands is disturbed in MCM2 HBD mutant cells. Thus, we successfully established cancer cell models with impaired histone inheritance.

Since parental histones are the carriers of histone PTMs through cell divisions, we explored the impact of impaired parental histone inheritance on histone modification profiles in MCM2 mutant cells. Globally, the levels of H3K27me3 and H3K4me1 increased in MCM2 mutant cells relative to their WT counterparts (Fig. 1c and Supplementary Fig. 1i). We further profiled the genome-wide distributions of these histone marks in MCM2-2A MCF-7 cells and observed distribution changes as well (Supplementary Fig. 2a, b). In addition, we also observed alterations of chromatin accessibility in MCM2-2A mutant MCF-7 cells (Supplementary Fig. 2b).

To gain insights into the altered patterns of histone marks, we annotated all identified peaks for each histone mark in MCF-7 cells. Downregulated H3K27me3 peaks were significantly enriched at promoters, 5′ UTRs, exons, transcription termination sites (TTSs), and CpG island regions in MCM2-2A mutant cells compared to the WT cells (Fig. 1d). While the upregulated H3K27me3 peaks were mainly distributed in intergenic regions similar to the stable peaks (unchanged), (Supplementary Fig. 2c, d and Supplementary Table 1). The putative regulatory genes of histone mark alternations were mainly related to development, differentiation, and proliferation (Supplementary Fig. 3). Furthermore, alterations in repressed histone PTMs (exemplified by the intergenic region between *CNGB3* and *CNBD1*) were clustered together, away from active histone PTMs (exemplified by *KDM5C* and *RCOR2* loci) (Fig. 1e, f), suggesting that the global changes between repressive and active histone marks are distinct. In summary, impaired histone inheritance leads to the reprogramming of multiple histone PTMs in the MCM2-2A cellular model.

It has been found that MCM2 facilitates the recycling of parental histones to lagging strands, whereas POLE3 promotes the recycling to leading strands in both *Saccharomyces cerevisiae* and mESCs[7,8]. To further confirm the roles of parental histone inheritance in cancer cells, we deleted POLE3 in MCF-7 cell line by CRISPR-mediated knockout (KO) to investigate its impact on epigenetic reprogramming (Supplementary Fig. 4). The H3K36me3 eSPAN signals in POLE3 KO MCF-7 cells displayed a weak bias toward the lagging strand (Supplementary Fig. 4a), indicating POLE3 KO also impairs the transfer of parental histones to leading strands in cancer cells. In POLE3 KO and MCM2-2A mutant MCF-7 cells, we observed consistent changes in terms of global H3K27me3 and H3K4me1 levels and H3K27me3 occupancy (Supplementary Fig. 4b–d). These results indicate that impaired transfer of parental histones caused by POLE3 deletion or MCM2-2A mutation can lead to similar epigenetic reprogramming in MCF-7 cells.

### Disturbing histone inheritance reprograms the pattern of H3K27me3 and derepresses development-related genes in cancer cells

Previous studies indicated that propagations of repressive marks, such as H3K27me3 and H3K9me3 rely on a "read and write" mechanism, namely the modified parental histones on the daughter strand serve as templates for new histones[1]. Consistently, we observed that Polycomb repressive complex 2 (PRC2) and Polycomb repressive complex 1 (PRC1) occupancy alterations correlated with H3K27me3 changes in MCM2-2A mutant MCF-7 cells (Supplementary Fig. 5a, b). It is possible that the repressive marks on the lagging strand fail to be restored due to a lack of templates, which leads to global landscape changes. However, more than half of H3K27me3 peaks remained stable in MCM2 mutant cells throughout multiple cell divisions (Supplementary Tables 1 and 2). This observation suggests that the local "read and write" propagation theory is insufficient to explain the changes of H3K27me3 pattern.

To explore the mechanisms regulating H3K27me3 restoration, we compared the original chromatin state at altered H3K27me3 regions in MCF-7 cells, and observed significant H3K36me3 enrichments flanking the H3K27me3 downregulated peaks resulting from MCM2-2A mutation

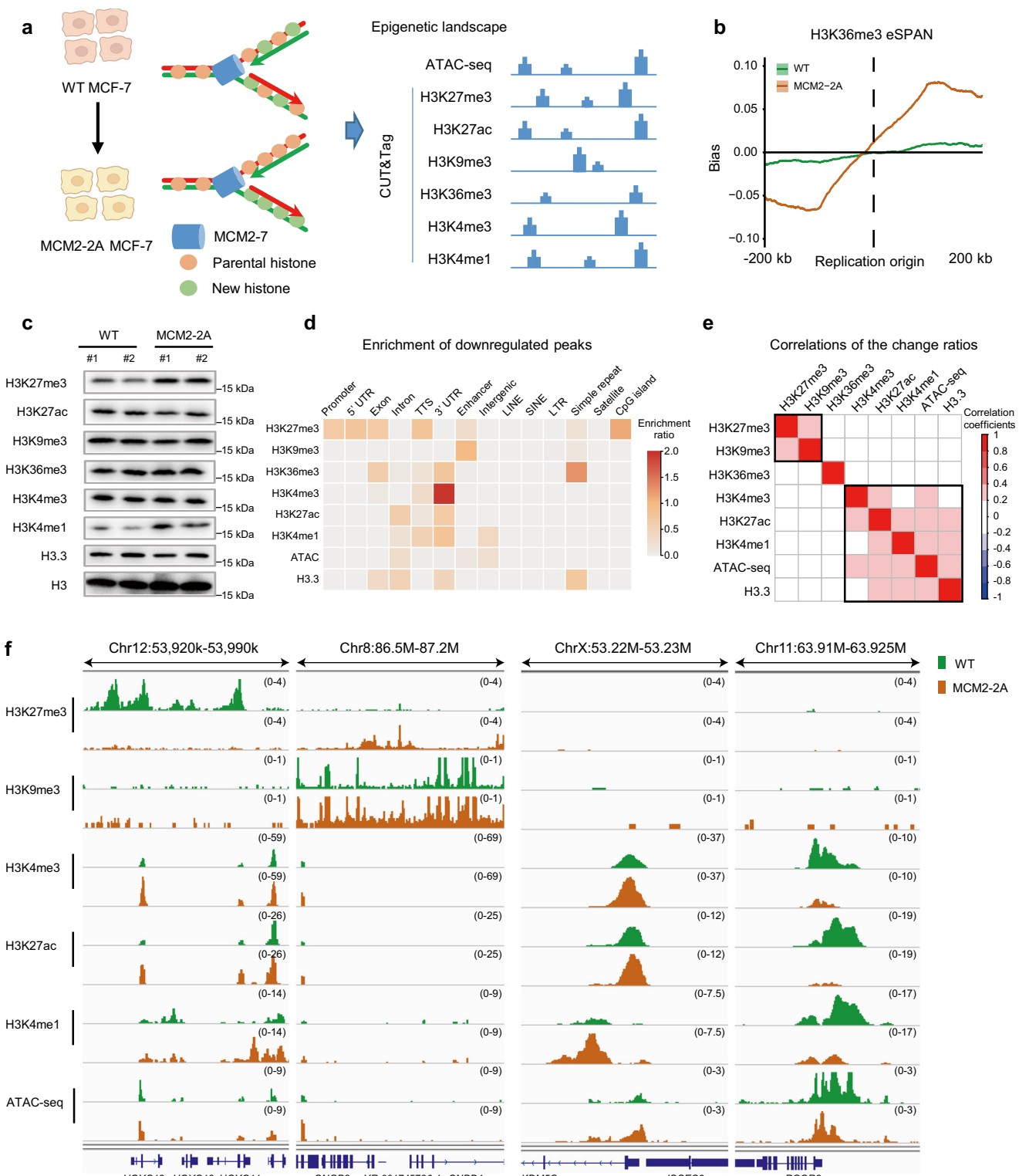

(Fig. 2a). The results imply that the H3K27me3 downregulated regions might locate within a more active chromatin context. Previous chromatin interaction studies have annotated two unique groups of genomic compartments, namely the open (A) and closed (B) types[29,30]. Therefore, we annotated the H3K27me3 peaks based on their genomic locations and found that the majority of upregulated H3K27me3 peaks were distributed in B-type compartments, whereas downregulated H3K27me3 peaks were mainly distributed in A-type compartments (Fig. 2b). Then, we compared the changes in H3K27me3 peaks among the three MCM2 mutant models, and found very similar pattern in the downregulated

H3K27me3 regions (Supplementary Fig. 6a). However, we observed highly similar H3K27me3 upregulated regions between MCF-7 and T47D cell lines, but not HEK293T (Supplementary Fig. 6b), suggesting a more complicated and undefined mechanism for H3K27me3 upregulation. In addition, the distribution of dysregulated H3K27me3 peaks in A- or B-type compartments in MCM2 mutant T47D cells was similar to that in MCM2 mutant MCF-7 cell line (Supplementary Fig. 6c). These results suggest that H3K27me3 tends to be lost in an active chromatin context and gained in a closed chromatin context in MCM2 mutant cancer cells.

**Fig. 1 | Impaired histone inheritance results in genome-wide epigenetic reprogramming in MCM2-2A mutant MCF-7 breast cancer cells. a** Experimental design: Epigenetic landscapes of chromatin accessibility and histone modifications were characterized using ATAC-seq and CUT&Tag in WT and MCM2-2A mutant MCF-7 cells. Created with BioRender.com. **b** Average bias of H3K36me3 in MCM2-2A mutant and WT MCF-7 cells, as determined using eSPAN. The exacerbated leading bias in MCM2-2A mutant cells means parental histone H3K36me3 recycled to the leading strand is more than that recycled to the lagging strand. **c** Western blot analysis of select histone marks and histone variant H3.3 using total cell lysates from MCM2-2A mutant and WT MCF-7 cells. Total histone H3 was used as a loading control. #1 and #2 indicate two independent clones in each group. This experiment was repeated 3 times independently with similar results. **d** Integration analysis showing which types of genomic regions are enriched for downregulated histone marks, histone variant H3.3 or chromatin accessibility (ATAC-seq) in MCM2-2A mutant MCF-7 cells. The color filled in each cell represents the enrichment ratio of the downregulated peaks superior to the stable peaks. The detailed calculation process for enrichment ratio is depicted in Methods. **e** Correlations of the change ratios between each pair of histone marks, histone variant H3.3 and chromatin accessibility. Briefly, the genome was divided into bins (10 kb each). FPKM signals were scanned and the intensity ratio (MCM2-2A mutant vs. WT FPKM intensity) was calculated for each bin, and then Pearson correlation coefficients for the intensity ratios were calculated for each pair. Correlation is displayed when $P < 0.01$. **f** Integrative Genomics Viewer tracks showing the distribution and correlation of histone modifications at indicated loci in MCM2-2A mutant and WT MCF-7 cells. In MCM2-2A mutant cells, H3K27me3 was downregulated at development-related *HOXC* cluster, while H3K27me3 and H3K9me3 were upregulated at intergenic regions between *CNGB3* and *CNBD1* (left two panels). The up- and downregulation of active histone marks (H3K4me3, H3K27ac), enhancer marker (H3K4me1), and chromatin accessibility (ATAC-seq) were consistent at *KDM5C* and *RCOR2* loci, respectively (right two panels). eSPAN enrichment and sequencing of protein-associated nascent DNA, UTR untranslated region, TTS transcription termination sites, LINE long interspersed nuclear elements, SINE short interspersed nuclear elements, LTR long terminal repeat retrotransposons.

---

Given the important regulatory function of H3K27me3, we hypothesized that the gene expression would subsequently change as well. We compared the transcriptome profile changes among three MCM2 mutant models and observed significantly overlapped up- and downregulated genes (Supplementary Fig. 7). We then explored the transcriptional changes of H3K27me3 target genes, defined by H3K27me3 enrichment at their promoters. As expected, the expression level of genes with downregulated H3K27me3 at promoters increased significantly, and the level of genes with upregulated or stable H3K27me3 remained low in MCM2 mutant cells (Fig. 2c and Supplementary Fig. 8a, b). Considering that upregulated H3K27me3 peaks were mainly distributed in the closed chromatin context (Fig. 2b), the transcriptional regulatory role of upregulated H3K27me3 should be limited. Hereafter, we focused on the H3K27me3 downregulated regions. Gene ontology (GO) term analysis indicates that these de-repressed genes are mainly involved in development and differentiation, such as gland development, epithelial cell differentiation, regulation of epithelial cell proliferation and epithelial to mesenchymal transition (EMT) (Fig. 2d). For example, within the term gland development, some of the key genes lost H3K27me3 at their promoters were significantly activated in MCM2-2A mutant MCF-7 cells (Fig. 2e). CUT&Tag-qPCR[31] and RT-qPCR further verified the association between H3K27me3 distribution and expression of growth- or EMT-related genes (Fig. 2f–h). Collectively, these findings suggest that, as a consequence of impaired histone inheritance, the H3K27me3 modifications flanked by transcriptionally active chromatin tend to be lost in MCM2-2A mutant cells, which leads to the activation of genes involved in development and differentiation (Supplementary Fig. 8c).

### Transcriptionally active chromatin remains stable despite disrupted inheritance of parental histones in MCM2-2A mutant cells

To track the change patterns of active histone PTMs, we investigated the original epigenetic profiles of the upregulated, stable, and downregulated H3K4me3 peaks in three MCM2 mutant cells and found that H3K4me3 signals at the stable regions were significantly higher than those at up- or downregulated regions (Fig. 3a and Supplementary Fig. 9a, b). Consistently, these H3K4me3 stable regions had higher chromatin accessibility (Fig. 3b), faster histone turnover (Supplementary Fig. 9c–e) and higher transcription activity marked by higher H3K36me3 levels (Supplementary Fig. 9f). The expression level of genes with H3K4me3 stable promoters was also higher than those with H3K4me3 up- or downregulated promoters in these three models (Fig. 3c and Supplementary Fig. 9g, h). We further examined other active histone marks in MCM2-2A MCF-7 cells and found a similar changing pattern for H3K27ac (Supplementary Fig. 10a–d), and a more

stable H3K4me1 signal at super-enhancers than repressed enhancers (Supplementary Fig. 10e, f). Collectively, these results indicate that, despite the disturbance in parental histone inheritance, transcriptionally active chromatin with fast histone turnover tends to remain stable through cell divisions, as exemplified by the histone marks at *GAPDH* locus (Fig. 3d). Transcription resumption and replication-independent histone exchange contribute to the active chromatin restoration, ensuring that transcriptionally active chromatin persists through cellular mitosis.

### Impaired histone inheritance promotes tumor growth and invasion in vivo

We observed H3K27me3 loss at the promoters of genes regulating mammary gland development, epithelial cell proliferation and EMT when parental histone inheritance is disturbed (Fig. 2d). These results suggest parental histone inheritance may participate in regulating proliferation and differentiation during tumor progression. We therefore transplanted barcoded WT or MCM2-2A mutant MCF-7 cells into immunocompromised mice (Fig. 4a), and strikingly, we found that the mutant tumors grew a lot faster than WT (Fig. 4b–d). Moreover, the overall survival of mice bearing MCM2-2A mutant tumors was significantly reduced (Fig. 4e). In the MCM2-2A mutant primary tumors, we observed cancer cells invaded into blood vessels and abdominal muscles and appeared outside the capsule (Fig. 4f). Furthermore, we detected lung metastasis in 6 mice (75%) bearing MCM2-2A mutant tumors, whereas only in 2 mice (25%) with WT tumors ($n = 8$ in each group, $P = 0.045$) (Fig. 4g). In addition, we cultured the tumor cells in organoids to mimic the in vivo environment[32] and found that the MCM2-2A and POLE3 KO MCF-7 organoids grew significantly faster than WT (Supplementary Fig. 11a). Similarly, the MCM2-90A T47D organoids also acquired growth advantages compared to WT (Supplementary Fig. 11b). The percentage of proliferating cells was higher in both MCM2-2A mutant and POLE3 KO MCF-7 organoids than WT (Supplementary Fig. 11c, d). We observed similar changes in gene expression between POLE3 KO and MCM2-2A mutant MCF-7 organoids (Supplementary Fig. 11e–g), and the changed genes were mainly involved in cell growth, proliferation, migration, mammary gland development and apoptotic process (Supplementary Fig. 11f), which is consistent with the accelerated growth phenotypes. These results denote that impairing parental histone inheritance promotes tumor growth and facilitates tumor cell metastasis.

### Histone inheritance disorder promotes tumor progression by forming distinct subclones

To characterize the evolution of MCM2-2A mutant tumor at single-cell level, we performed scRNA-seq on tumor samples collected 4 or 7 weeks post orthotopic transplantation. As expected, impaired histone

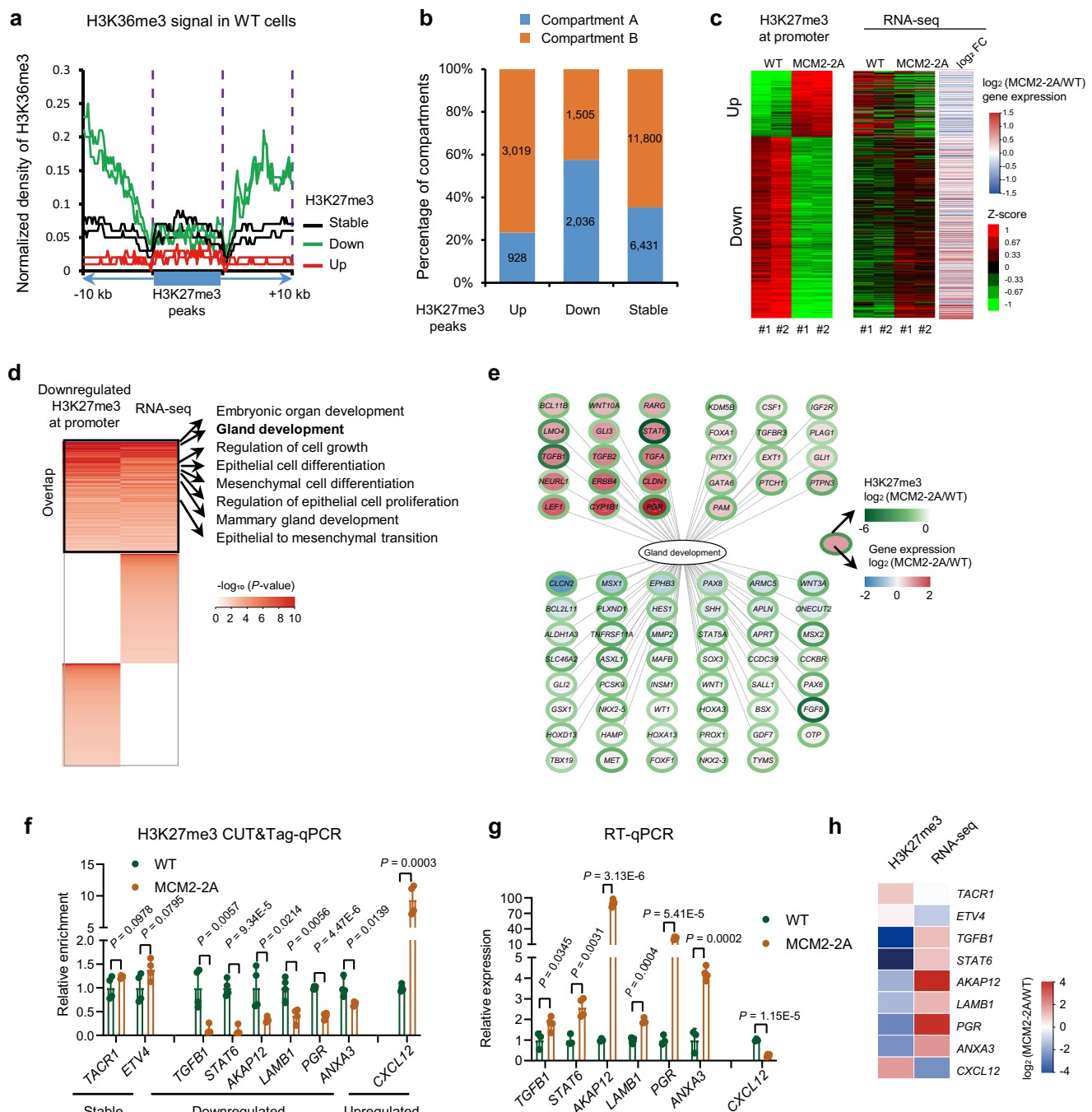

**Fig. 2 | MCM2-2A mutation leads to H3K27me3 reprogramming and derepression of development-related genes in MCF-7 cells. a** Average signal of H3K36me3 in WT MCF-7 cells at peaks (±10 kb) at which H3K27me3 was upregulated, stable, or downregulated in MCM2-2A mutant vs. WT cells. In WT cells, levels of H3K36me3 in areas flanking H3K27me3 downregulated peaks were higher than that flanking H3K27me3 stable or upregulated peaks. Two independent clones were shown. **b** Proportion of H3K27me3 peaks belonging to A-type (open) vs. B-type (closed) compartments, stratified by whether the peaks were upregulated, stable, or downregulated in MCM2-2A mutant vs. WT MCF-7 cells. The A/B compartment profiles were obtained from the published normalized Hi-C matrix data of MCF-7 (GEO accession: GSE66733)[29]. **c** Heatmap showing signal intensity differences in H3K27me3 at promoters (left panel) and corresponding changes of gene expression (right panel) in MCM2-2A mutant vs. WT MCF-7 cells. **d** Heatmap showing *P*-values from a GO analysis of (1) genes with downregulated H3K27me3 at their promoters, as detected by CUT&Tag, and (2) differentially expressed genes between MCM2-2A mutant and WT cells, as detected using RNA-seq. GO terms of

these gene sets overlapped well. One-sided hypergeometric test without adjustment was used to calculate statistical significance. **e** Fold change in expression levels (MCM2-2A mutant/WT) of genes associated with gland development at whose promoter H3K27me3 was downregulated in MCM2-2A mutant vs. WT MCF-7 cells. The color of each gene's circular border represents the fold change in H3K27me3 levels at its promoter, whereas the color inside represents the fold change in gene expression. **f** CUT&Tag-qPCR for H3K27me3 occupancy at selected regions in MCM2-2A mutant and WT MCF-7 cells. Data are presented as mean values ± SD. n = 4 (2 experiments over 2 independent clones). Two-sided Student's *t* test. **g** mRNA levels measured by RT-qPCR in MCM2-2A mutant and WT MCF-7 cells. Data are presented as mean values ± SD. WT, *n* = 3 independent clones, MCM2-2A, *n* = 4 (2 experiments over 2 independent clones). Two-sided Student's *t* test. **h** Heatmap showing the fold change of H3K27me3 signal and RNA expression identified by CUT&Tag sequencing and RNA-seq, respectively. FC fold change, qPCR quantitative real-time PCR, RT reverse transcription.

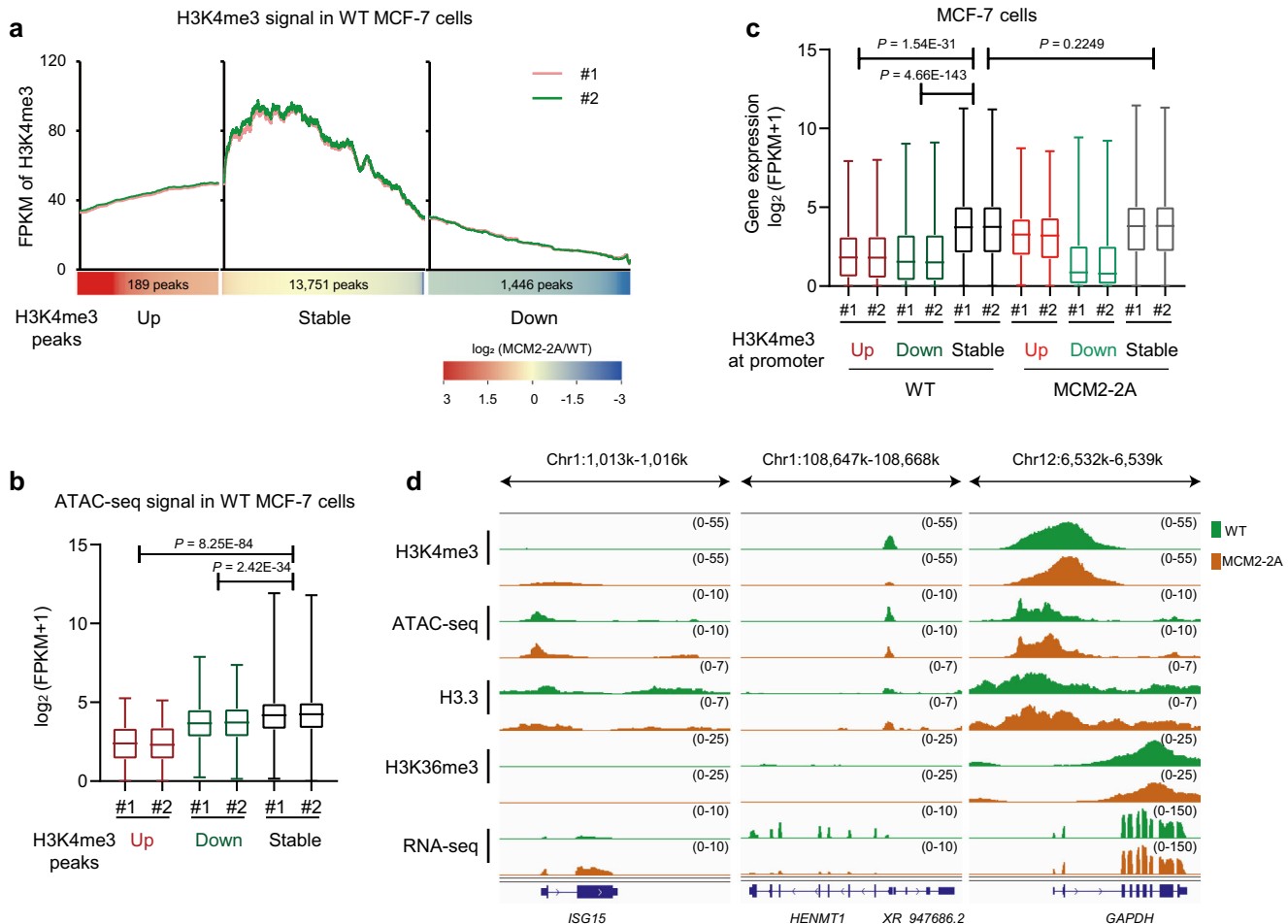

**Fig. 3 | Transcriptionally active chromatin state remains stable during the epigenetic reprogramming of MCM2-2A mutant MCF-7 cells. a** H3K4me3 signal in WT MCF-7 cells, at H3K4me3 peaks that were upregulated, stable, or downregulated in MCM2-2A mutant vs. WT cells. #1 and #2 indicate two independent WT MCF-7 clones. **b** Boxplots depicting chromatin accessibility (ATAC-seq signal) in WT MCF-7 cells at H3K4me3 peaks upregulated (*n* = 189), stable (*n* = 13,751), or downregulated (*n* = 1446) in MCM2-2A mutant vs. WT cells. Two-way repeated measures ANOVA adjusted by LSD for multiple comparisons indicates that the chromatin accessibility at H3K4me3 stable regions was significantly higher than that at H3K4me3 up- or downregulated regions. **c** Boxplots representing expression (RNA-seq data) of genes with H3K4me3 upregulated (*n* = 249), stable (*n* = 18,508), or downregulated (*n* = 1151) promoters in MCM2-2A mutant (right) and WT (left) MCF-7 cells. Two-way repeated measures ANOVA adjusted by LSD for

multiple comparisons indicates that the expression of genes in WT cells with H3K4me3 stable promoters was significantly higher than that with H3K4me3 up- or downregulated promoters, in addition, the expression of genes with H3K4me3 stable promoters was similar between MCM2-2A mutant and WT cells. The box plots in **b**, **c** display the median, upper and lower quartiles; the whiskers show 1.5× interquartile range (IQR). **d** Integrative Genomics Viewer tracks showing distributions of histone modifications and histone variant H3.3 (CUT&Tag data), chromatin accessibility (ATAC-seq data), and gene expression (RNA-seq data) for selected loci in MCM2-2A mutant and WT MCF-7 cells. The chromatin state for *GAPDH* (right panel) was more active than *ISG15* with upregulated H3K4me3 (left panel) and *HENMT1* with downregulated H3K4me3 (middle panel) in WT cells and remained stable in MCM2-2A mutant cells. FPKM, Fragments Per Kilobase per Million mapped fragments.

inheritance induced dramatic gene expression changes, indicated by substantial differences in cell clustering between WT and MCM2-2A tumors (Supplementary Fig. 12a). We then analyzed the expression of key breast cancer genes[33,34], and found that both WT and mutant cancer cells kept the basic characteristics of the MCF-7 cell line. For example, both tumors were mainly composed of epithelial breast cancer cells (*EPCAM*+) with a similar distribution of hormone receptors (*ERBB2* for HER2, *ESR1* for estrogen receptor, and *PGR* for progesterone receptor) and histology markers (*CDH1* for E-cadherin) (Supplementary Fig. 12b, c). These cells were largely luminal (*KRT8*+, *KRT18*+) rather than basal (*KRT5*−, *KRT14*−), which matches well with the typical expression pattern of MCF-7[35] (Supplementary Fig. 12c). These results support our finding that transcriptionally active chromatin remained stable in these mutant cells.

To further explore the transcriptional diversity within the tumor, we resolved the tumor cells into five clusters based on their distinct transcriptome profiles (Fig. 5a and Supplementary Fig. 12d, e). Clusters

2 and 5 were almost entirely composed of MCM2-2A mutant cells, whereas cluster 3 was entirely composed of WT cells (Fig. 5b). Through comparing tumors collected at different time points, we found that the cell composition in each cluster within mutant tumor changed dramatically at late time point (7 weeks), whereas remained stable within WT tumors (Fig. 5c). Notably, the mutant-specific cluster 2 gradually became the dominant cluster at late time point (Fig. 5c). We also observed that cells within cluster 2 expressed high levels of genes regulating cell growth, response to hormone, epithelial cell differentiation and gland development (Fig. 5d, e and Supplementary Fig. 12f), supporting the growth advantage phenotype. Cells in the mutant-specific cluster 5 expressed high levels of genes associated with DNA replication, mitotic cell cycle, and cell proliferation-related pathways (G2M checkpoint, E2F targets, MYC targets V1 & V2, and mitotic spindle), indicating a fast-cycling phenotype (Fig. 5d and Supplementary Fig. 12g). Besides, we obtained the dysregulated gene sets that are significantly associated with invasion, metastasis or

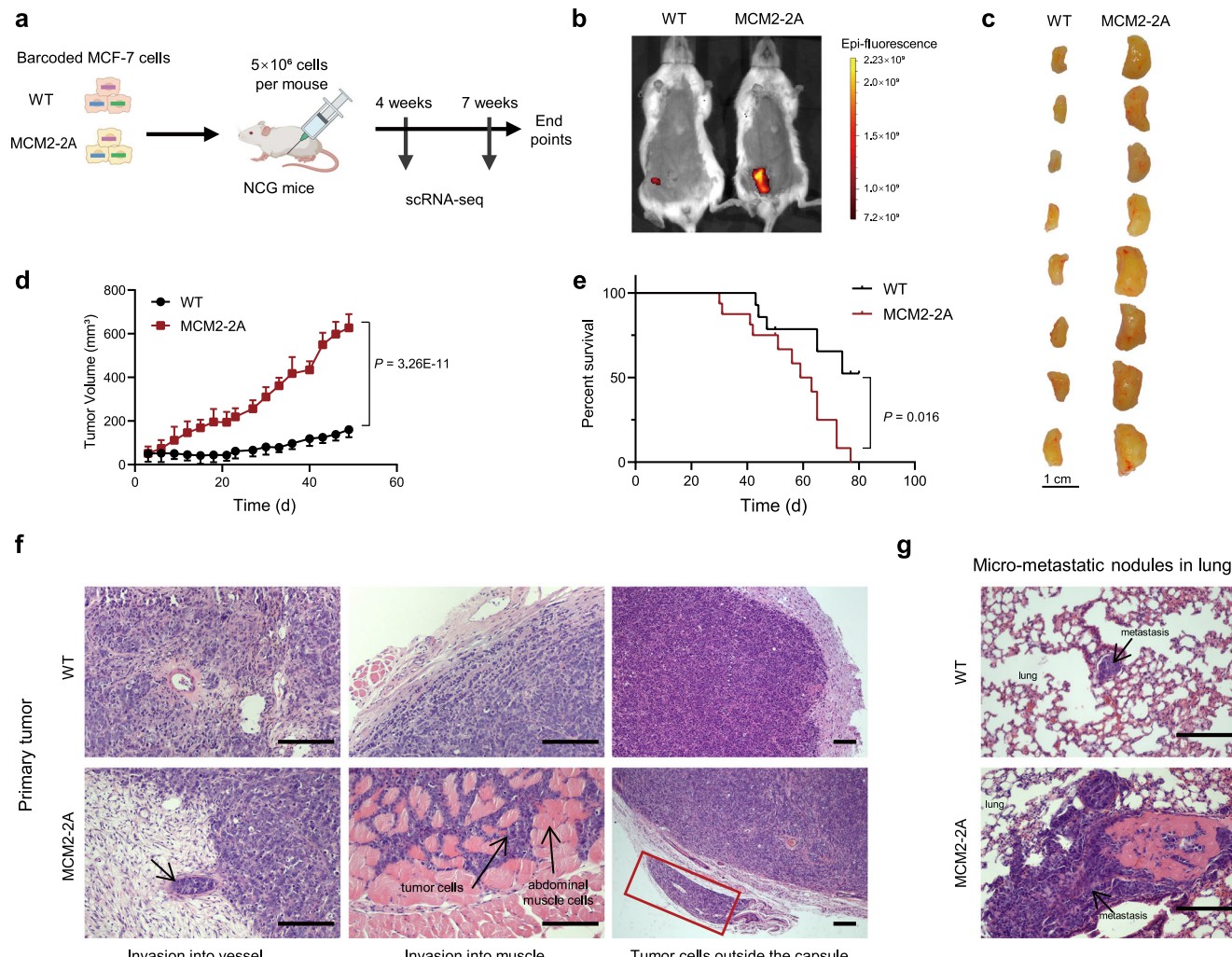

**Fig. 4 | Impaired parental histone inheritance facilitates tumor growth and metastasis in vivo. a** Experimental design: MCM2-2A mutant and WT MCF-7 cells with unique barcodes were expanded and transplanted into the fourth mammary fat pads of immunocompromised NOD/ShiLtJGpt-Prkdc[em26Cd52]Il2rg[em26Cd22]/Gpt (NCG) mice. Resulting tumors were subjected to scRNA-seq 4 and 7 weeks post-transplantation. Created with BioRender.com. **b** Representative IVIS eGFP fluorescence imaging of orthotopic MCM2-2A mutant and WT MCF-7 tumors at 7 weeks post-transplantation. **c** Representative images taken 7 weeks post-transplantation are shown for MCM2-2A mutant and WT MCF-7 tumors. **d** Average growth curves of tumors resulting from MCM2-2A mutant and WT MCF-7 cells. Error bars, SD ($n = 10$, each group); Two-sided Student's $t$ test. **e** Kaplan–Meier survival analysis of mice bearing MCM2-2A mutant and WT MCF-7 tumors (WT, $n = 15$; MCM2-2A, $n = 16$). $P = 0.016$, log-rank test. **f** HE staining showing the representative images of primary tumor tissue from MCM2-2A mutant and WT MCF-7 tumors. Scale bar, 200 μm. **g** HE staining showing the representative images of micro-metastatic nodules in the lung of mice bearing MCM2-2A mutant and WT MCF-7 tumors. Scale bar, 200 μm. The experiments in **f**, **g** were repeated in 8 mice independently with similar results.

prognosis in breast cancer patients from the GSEA website and analyzed the GSVA score of these gene sets in each cell. Cells in cluster 5 owned a higher score, suggesting an aggressive phenotype (Supplementary Fig. 12g). Moreover, we characterized the breast cancer function state of our single cells based on CancerSEA[36], which portrays human cancer single-cell functional state atlas. The cells in cluster 5 got higher scores of function states involving cell cycle, EMT, invasion and proliferation (Supplementary Fig. 12h), which further supports the functional importance of this cluster in tumor progression. To evaluate the recurring gene expression features, we classified our single cell RNA-seq data into luminal A, luminal B, HER2-enriched, and basal-like subtypes based on their transcriptome profiles using the established PAM50 method[37] (Supplementary Fig. 12i). The proportion of HER2-enriched and basal-like cells was quite low, which is consistent with the identified signature genes (Supplementary Fig. 12c). Nevertheless, we observed a significantly higher proportion of luminal B subtype in cluster 5. The prognosis of breast cancer patients with luminal B subtype is worse than those with luminal A subtype[38]. This result suggests

that MCM2-2A mutant cancer cells are more likely to evolve into a worse subtype. In summary, our findings indicate that impaired histone inheritance evoked by MCM2-2A mutation promotes tumor growth and invasion by facilitating the emergence of distinct subclones.

We further analyzed the correlation between in vitro CUT&Tag data and in vivo scRNA-seq data. We screened out the promoters with downregulated H3K27me3 or dysregulated H3K4me3 in MCM2-2A mutant MCF-7 cells, whose target genes are related to tumor growth, proliferation and metastasis, and explored their impact on gene expression in vivo (Fig. 5f, g). We found that 35% of the genes derepressed by H3K27me3 loss continued to be upregulated at 4 weeks post-transplantation. Additionally, most genes with dysregulated H3K4me3 maintained their changed expression patterns at 4 weeks post-transplantation. Furthermore, the correlation between the dysregulated histone marks and their target proliferation-related genes in MCM2-2A mutant and POLE3 KO MCF-7 organoids further suggests the important role of impaired histone inheritance in tumor growth

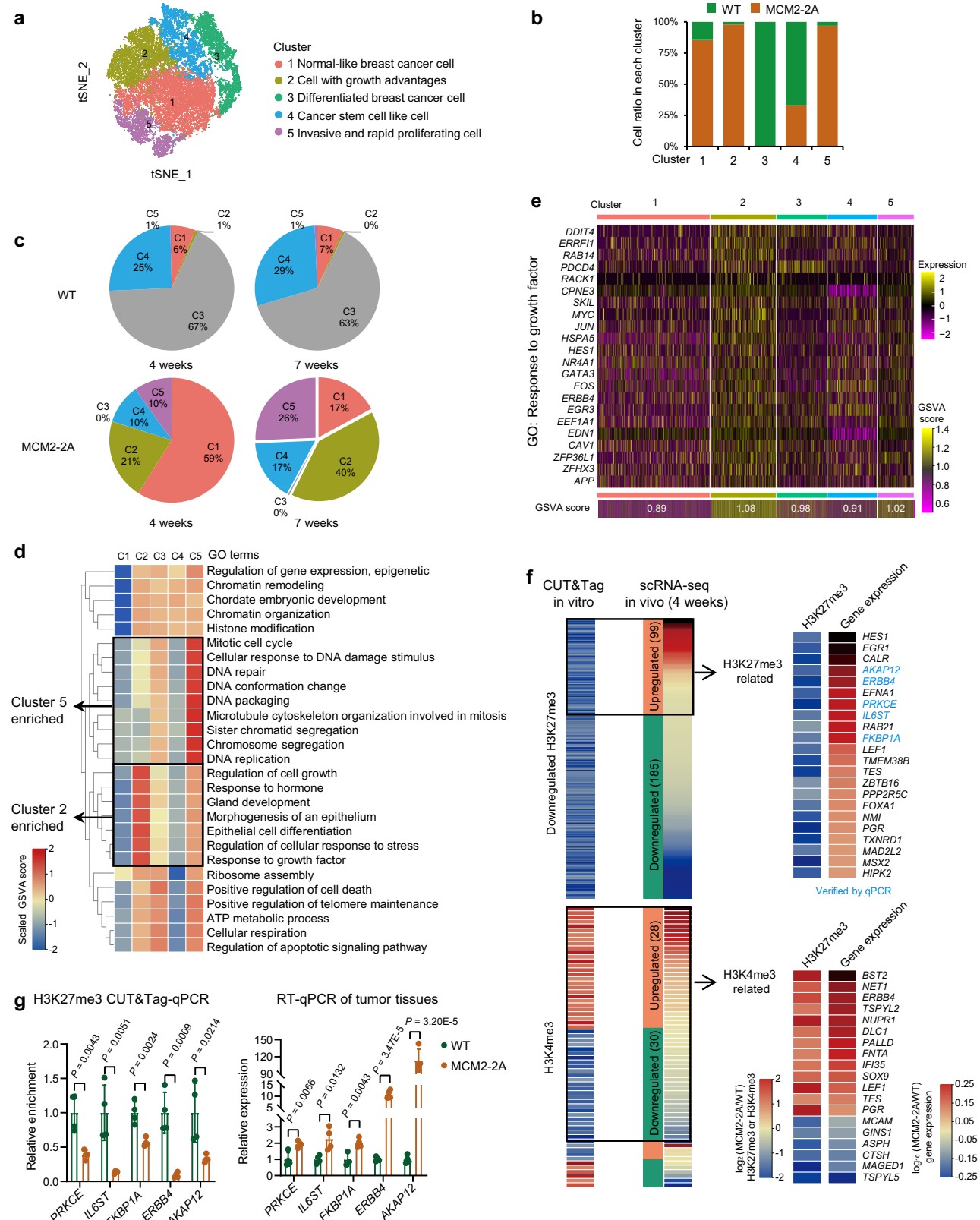

(Supplementary Fig. 13a, b). All the findings indicate that impaired histone inheritance promotes tumor progression through epigenetic reprogramming.

We also assessed the correlation between the expression of the upregulated genes in tumors derepressed by H3K27me3 loss and survival in breast cancer patients without endocrine- or chemo- treatment. We found that the upregulation of a proportion of these derepressed genes was associated with poor prognosis in patients (Supplementary Fig. 13c). Moreover, a proportion of dysregulated genes in tumors targeted by dysregulated H3K4me3 were also related to poor prognosis (Supplementary Fig. 13d). These results suggest that the dysregulation of genes targeted by the impaired H3K27me3 or

**Fig. 5 | Impaired histone inheritance in MCM2-2A mutant MCF-7 cells promotes tumor progression by forming distinct subclones in vivo. a** Visualization of scRNA-seq data showing that cells in MCM2-2A mutant and WT tumors fell into five distinct clusters. scRNA-seq data from tumors analyzed 4 and 7 weeks post-transplantation were projected onto a *t*-distributed stochastic neighbor embedding (*t*-SNE) plot. Five clusters were identified and characterized. **b** Proportion of MCM2-2A mutant and WT tumor cells in each of the five clusters. **c** Proportion of cells from each cluster in WT (upper two panels) and MCM2-2A mutant (lower two panels) tumors harvested at 4 weeks (left two panels) and 7 weeks (right two panels) post-transplantation. **d** Heatmap showing top enriched GO terms for each cluster. The color filled in each cell represents the average gene set variation analysis (GSVA) score for that term in that cluster. **e** Heatmap of expression for genes associated with the response to growth factor GO term in each cell from five clusters, with GSVA scores of this gene set shown at the bottom. Each column represents a single cell. **f** Heatmap showing the changes of H3K27me3 (upper panel) and H3K4me3 (lower panel) signal at promoters and the expression changes of their target genes post-transplantation. Growth-, proliferation- or metastasis-related genes with H3K27me3 downregulated or H3K4me3 dysregulated promoters were displayed. The fold change of RNA expression is calculated by comparing the average expression of each single cell in MCM2-2A mutant to WT tumor collected 4 weeks post-transplantation. **g** CUT&Tag-qPCR for H3K27me3 occupancy at the promoter regions of selected genes (blue) in **f**, and corresponding gene expression measured by RT-qPCR in MCM2-2A mutant and WT MCF-7 tumors. Data are presented as mean values ± SD. CUT&Tag-qPCR, *n* = 4 (2 experiments over 2 independent clones). RT-qPCR, *n* = 4 tumors from independent mice. Two-sided Student's *t* test. GO Gene Ontology, qPCR quantitative real-time PCR, RT reverse transcription.

H3K4me3 distribution is associated with poor prognosis in breast cancer patients.

### Impaired histone inheritance drives tumor evolution by forming neo-clones with fitness advantage

The complex microenvironment in mammary gland such as hormones stimuli, cancer-associated fibroblasts, macrophages, and hypoxia, can influence tumor cell evolution. To investigate the clonal evolution of MCM2-2A mutant cells in vivo, we reconstructed clonal relationships based on the lineage and RNA recovery (LARRY) lentiviral barcodes using the scRNA-seq data collected 4 or 7 weeks post-transplantation. We found that in the MCM2-2A mutant tumors, dominant clones gradually took over MCM2-2A mutant tumors by 7 weeks post-transplantation (Fig. 6a). In addition, the number and size of dominant clones were greater in MCM2-2A mutant tumors compared to the WT (Supplementary Table 3). These data indicate that MCM2-2A mutant cancer cells are more likely to form dominant clones compared to WT cells in vivo. We further found that the cells composing dominant clones were mainly located in cluster 2 (Fig. 6b and Supplementary Fig. 14a). The marker genes of dominant clones were related to the regulation of epithelial cell proliferation (Supplementary Fig. 14b–e), implying a growth advantage. These genes were also expressed at the highest levels in the cells within cluster 2 (Supplementary Fig. 14f). These results indicate that the dominant clones with growth advantages in MCM2-2A mutant tumors were mainly newly emerged clones from cluster 2.

To further investigate the mechanism of tumor evolution, we analyzed in vitro scRNA-seq data of WT and MCM2-2A mutant MCF-7 cells. Based on their transcriptome profiles, we resolved the cells into five clusters with two clusters in WT cells and three clusters in MCM2-2A mutant cells (Fig. 6c), suggesting higher transcriptional diversity of MCM2-2A mutant cells. Notably, cluster 2 cells expressed high level of genes regulating DNA replication, mitotic cell cycle and cell division, indicating growth advantages (Fig. 6d). Of note, a few proliferation-promoting genes derepressed by H3K27me3 loss were highly expressed in cluster 2, such as *POP7*[39], *TNNT1*[40] and *TYMS*[41]. Moreover, cluster 2 cells also expressed high levels of genes involved in chromatin remodeling and oxidative phosphorylation (Fig. 6d). It has been shown that increased oxidative phosphorylation is critical for lung metastasis in the human breast cancer[42]. In addition, cells in cluster 4 expressed high levels of genes regulating epithelial cell migration and differentiation (Supplementary Fig. 14g), which could potentially also promote lung metastasis. We then reconstructed the lineage relationships between the cells in vitro and the tumor cells collected 4 weeks post-transplantation based on their LARRY barcodes (Fig. 6e). We annotated the cells that would grow into dominant clones in MCM2-2A mutant tumors, and found that they mainly originated from cluster 2 in vitro (Fig. 6f). Given the function of highly expressed genes in cluster 2 in vitro, we speculate that some MCM2-2A mutant cells might have already obtained the ability of accelerated proliferation in vitro.

All these results support our model whereby impaired inheritance of parental histones could derepress genes involved in proliferation and development and hence generate cells that express high levels of proliferation-related genes, which could favor advantageous clone formation and drive tumor growth.

## Discussion

In this study, we found that disturbing the inheritance of parental histones during DNA replication could reprogram the epigenetic landscape of cancer cells and facilitate tumor growth and metastasis. MCM2-2A mutation disrupts the transfer of parental histones with their associated PTMs to the lagging DNA strand in replicating cancer cells. Given the slow H3K27me3 restoration rate from replication-coupled dilution[43], H3K27me3 modifications were gradually lost through multiple cell divisions. The genes deprived of H3K27me3 at their promoters are poised for transcription and could provide tumor cells with greater plasticity in adapting to external stimuli. In the complex microenvironment of mammary gland, MCM2-2A mutant cells might generate a variety of newly emerged clones, among which the most advantageous ones will take over as tumor progresses (Supplementary Fig. 14h).

In the MCM2-2A mutant MCF-7 cells, although the transfer of parental histones to the lagging strand was blocked during replication, more than half of H3K27me3 remained stable (Supplementary Table 2). This result suggests that the restoration of H3K27me3 in MCF-7 cells is not solely dependent on the inheritance of parental histones and read-and-write propagation. In recent years, it has been discovered that DNA sequence, DNA methylation and chromatin high-order structure could also contribute to H3K27me3 restoration[44]. The accurate recycling of H2AK119ub1 during DNA replication, which is not disturbed in MCM2-2A mutant cells, could facilitate PRC2 recruitment and guide the accurate restoration of H3K27me3[45]. PRC2 could re-establish H3K27me3 pattern de novo with high fidelity genome-wide in mESCs[46], but the recovery of H3K27me3 patterns is incomplete in the neural progenitor cells (NPCs) and immortalized mouse embryonic fibroblasts (iMEFs)[47]. Therefore, the inheritance of parental histones carrying H3K27me3 is necessary for maintaining H3K27me3 at specific regions in differentiated cells. Besides, our study indicates that transcription plays a vital role in reprogramming H3K27me3 landscape in MCM2-2A mutant cancer cells. Transcriptional memory hinges on the double-negative feedback between Polycomb-mediated silencing and active transcription, and its activation is a widespread developmental feature of PRC2 target genes that emerges during cell differentiation[47]. Loss of H3K27me3 in MCM2-2A mutant cancer cells appears to allow the nearby transcriptionally active state to invade formerly repressive genomic loci and activate the correspondent gene expression. Subsequently, active transcription at these aberrantly activated loci would antagonize Polycomb activity and be sustained in a transcription-dependent way. The failure of H3K27me3 restoration across cell divisions can re-activate mammary gland development processes (Fig. 2d)

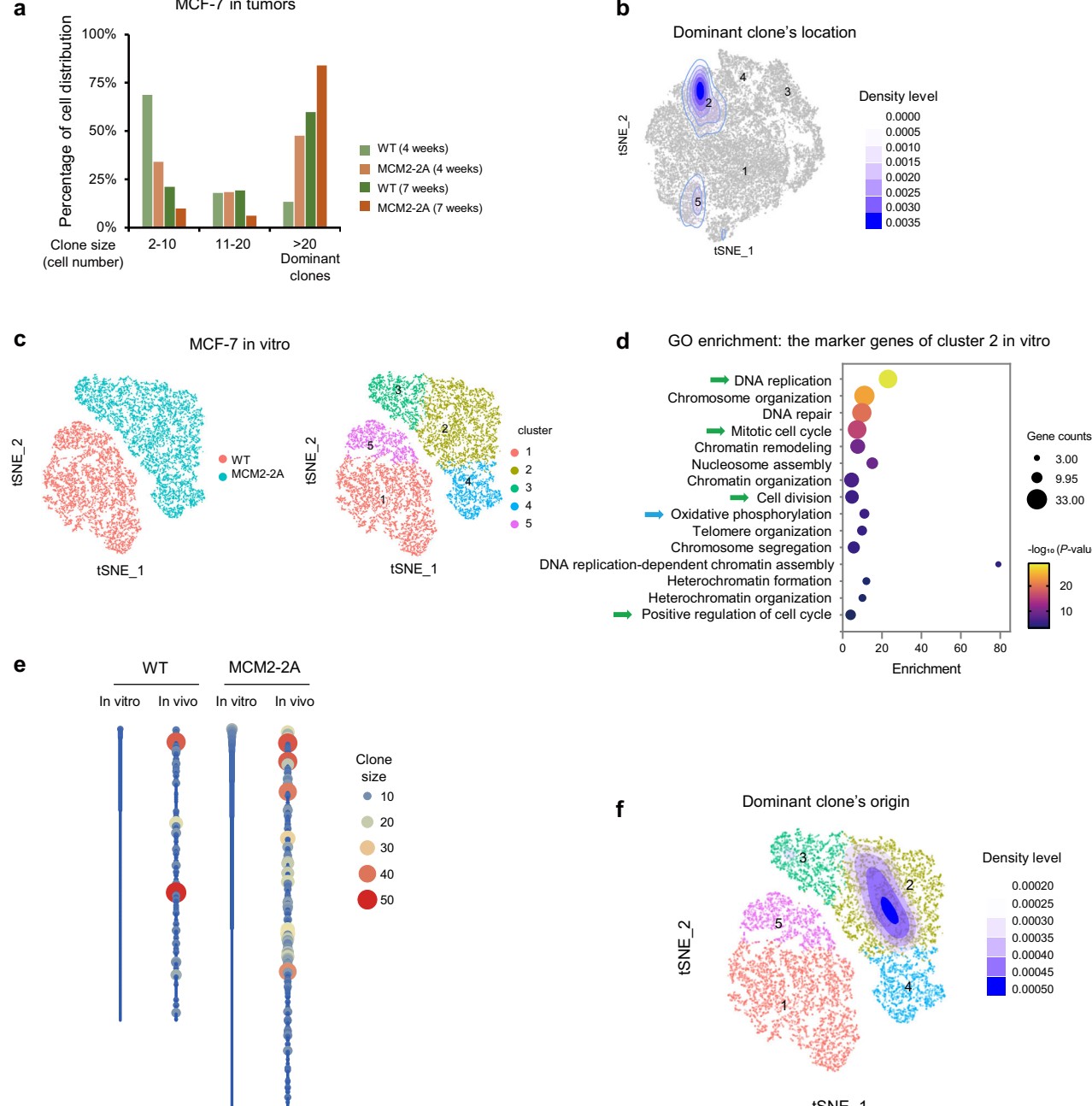

**Fig. 6 | Histone inheritance disorder drives dominant clone formation in MCM2-2A mutant MCF-7 tumors. a** Percentage of cells distributed in clones containing 2–10, 11–20, or >20 cells in MCM2-2A mutant and WT tumors. Cells harvested from mouse tumors 4 or 7 weeks post-transplantation are shown, respectively. Cells with the same lineage barcode are defined as a "clone" from the same ancestor cell. **b** Contour plots showing cell density of a representative dominant clone projected onto *t*-SNE plot for MCM2-2A mutant cells. Cells in the dominant clone mainly distribute in cluster 2. **c** scRNA-seq of in vitro MCM2-2A mutant and WT MCF-7 cells. Cells were dissolved into 5 clusters. Clusters 2, 3 and 4 were mainly composed of MCM2-2A cells, and clusters 1 and 5 were composed of WT cells. **d** Gene ontology enrichment analysis for marker genes of cluster 2 in vitro. One-sided hypergeometric test without adjustment was used to calculate statistical significance. Green arrows represent cell proliferation and blue arrow represents metabolism reprogramming. **e** Barcode abundance (clone size) in the tumor cell population prior to engraftment (in vitro) and in tumors 4 weeks post-transplantation (in vivo). Barcodes are ordered vertically according to their initial abundance (highest to lowest from top to bottom) in vitro, only the barcodes that are simultaneously captured both in vitro and in vivo are shown. The number of clones that would grow into dominant clones from MCM2-2A MCF-7 cells is far more than that from WT cells. **f** Contour plots projected onto *t*-SNE plot showing cell density of in vitro MCM2-2A mutant cells that would grow into dominant clones. *t*-SNE *t*-distributed stochastic neighbor embedding.

that are often hijacked by breast cancer cells as drivers for tumor progression[48,49].

MCM2-2A mutation has been reported only disrupting the histone chaperone function of MCM2 without affecting its role in DNA replication[26]. Defects in MCM2's histone chaperone function is not expected to promote cell proliferation[4,6]. Unexpectedly, we found that

MCM2-2A mutant tumors grew faster than WT in vivo, with significantly changed H3K27me3 pattern. H3K27me3 reprogramming has been observed in various cancer types[50–52]. Loss of H3K27me3 generates an overly permissive chromatin, which allows epigenetic plasticity, whereas gain of H3K27me3 results in overly restrictive chromatin and tumor suppressor genes repression[3]. Both scenario

could confer a fitness advantage and contribute to tumorigenesis[3]. For example, loss of H3K27me3 at the promoter of *TGFA*, a growth factor that regulates the autocrine growth of breast cancer cells[53], leads to active gene expression (Fig. 2e). While at the promoter of endothelin B receptor (*EDNRB*), a candidate tumor suppressor[54], the gain of H3K27me3 stringently represses the gene expression (Supplementary Fig. 2d). Both transcription changes are expected to contribute to the evolution of MCM2-2A mutant tumors. As a consequence of long-time selection, it is very difficult to screen out the exact candidate genes that change their epigenetic state in vitro and drive tumor progression in vivo. However, the target genes of downregulated H3K27me3 are enriched in GO terms regulation of cell growth, epithelial cell differentiation and gland development (Fig. 2d), similar to the GO terms that the marker genes of cells within in vivo cluster 2 are enriched (Fig. 5d). These results imply that the reprogramming of H3K27me3 derived from impaired histone inheritance might drive rapid tumor proliferation. Although the influence of impaired histone inheritance on the transcriptionally active regions is minor than repressive regions, we observed the persisting effects of dysregulated H3K4me3 on gene expression post-transplantation, suggesting an important role of H3K4me3 reprogramming in tumor progression. We focused on H3K27me3 reprogramming in this article, while the dramatic changes of enhancers and H3K9me3 marked heterochromatin regions could also participate in driving tumor progression.

Epigenomic reprogramming is one of the dominant forces in the development of tumor metastasis[55–57]. Timed addition and removal of H3K27me3 is critical for enabling proper differentiation throughout development but improperly modifications can lead to inappropriate cellular phenotype transformation[58]. As we observed, the reprogramming of H3K27me3 invoked by the dysregulated histone inheritance is related to mesenchymal cell differentiation and epithelial to mesenchymal transition (Fig. 2d and Supplementary Fig. 3), which could promote breast cancer metastasis. In the current study, our model revealed that cancer cells would evolve faster when parental histone inheritance are disturbed. This is the direct evidence showing impaired histone inheritance could promote tumor evolution.

In conclusion, our research reveals that impaired histones inheritance during DNA replication promotes tumor growth and metastasis. It provides another insight into how epigenetic instability leads to tumor progression. As epigenetic plasticity plays an important role in therapy resistance and drug tolerance, future studies could pay more attention to the variable regions resulting from impaired histones inheritance. In addition, this research puts forward a conception that treatment targeting the abnormal epigenetic inheritance might improve the patients' outcome by maintaining the epigenetic stability.

## Methods

### Cell culture
Regular MCF-7, T47D and HEK293T cell lines were obtained from ATCC. These cell lines were cultured using Dulbecco's modified Eagle's medium (DMEM) containing 10% fetal calf serum, 100 units/ml penicillin, 100 mg/ml streptomycin, and 2 mM L-glutamine.

### Genome editing
CRISPRCas9–guided gene editing was performed as according to Zhang's protocol[59]. Briefly, gene editing was performed to disable the histone-binding domain of MCM2 by mutating the tyrosine (Y81 and Y90) to alanine (A) and knockout POLE3. Oligos (Supplementary Table 4) were synthesized and inserted into vector pX458. Cells were transfected with the pX458 vector and a single-stranded oligonucleotide donor (Supplementary Table 4) to mutate MCM2 or pX458 vector only to delete POLE3 using the Celetrix electroporation kit. GFP-positive single cells were sorted 48 h post-transfection by BD FACSAria III and cultured for 4 days. After assessing targeted modification efficiency, cells were isolated by diluting and cultured for 2–3 weeks.

Individual clones were expanded for genotyping. Genetic mutations were verified using Sanger sequencing. Two MCM2-2A mutant MCF-7 clones, one MCM2-2A HEK293T clone, one MCM2-90A T47D clone, and one POLE3 KO MCF-7 clone were obtained.

### Preparation of cell extracts and Western blotting analysis
Cells were lysed in RIPA buffer (Cell Signaling Technology, Cat.#9806) and boiled for 5 min after adding the SDS-loading buffer. For Western blotting analysis, antibodies against MCM2 (Cell Signaling Technology, Cat#3619), β-Actin (Beyotime, Cat.#AF0003), POLE3 (Bethyl, Cat.#A301245A), H3K36me3 (Active Motif, Cat.#61021), H3K27me3 (Cell Signaling Technology, Cat.#9733), H3K27ac (Cell Signaling Technology, Cat.#8173), H3K9me3 (Abcam, Cat.# ab8898), H3K4me3 (Active Motif, Cat.#39159), H3K4me1 (Cell Signaling Technology, Cat.#5326), H3.3 (proteintech, Cat.#13754-1-ap) and H3 (Abcam, Cat.#ab1791) were used in the dilution of 1:1000. HRP-conjugated goat anti-mice (Beyotime, Cat.#A0216) or HRP-conjugated goat anti-rabbit (Beyotime, Cat.#A0208) antibody was incubated in the dilution of 1:5000 according to the origin of the primary antibody. The bands were visualized using Tanon App for Biology Software (Version 1.0.0000) after incubating with ECL mix (Beyotime, Cat.#P0018FM). Full scan blots, see the Source Data file. Relative quantification analysis of Western blot band intensity was performed using ImageJ (v150).

### RNA-seq and ATAC-seq
Total RNA was extracted using TRIzol reagent (Invitrogen, Cat.#15596026CN). RNA-seq library preparation and deep sequencing were performed by Annoroad Gene Technology Co., Ltd. ATAC-seq was performed according to Greenleaf's protocol[60]. Briefly, $5 \times 10^4$ fresh cells were taken for ATAC-seq. After once wash with cold PBS, the cells were resuspended with 50 μl lysis buffer and incubated on ice for 3 min. After once wash with cold PBS, the cells were resuspended with 50 μl transposition reaction buffer containing 1 μl Tn5 Transposase (Novo protein, Cat.#M045) and shaken at 37 °C for 30 min. Then the library PCR amplification was performed after DNA purification. Two independent clones for each group were sequenced.

### RT-qPCR
Total RNA was extracted using TRIzol reagent according to the manufacturer's instructions. cDNA was generated using 1 μg RNA by PrimeScript™ RT reagent Kit with gDNA Eraser (Takara, Cat.#RR047Q). Quantitative real-time PCRs (qRT-PCR) were performed using SYBR Green Premix Pro Taq HS qPCR Kit (Accurate biology, Cat.#AG11701) by qTOWER 3 or Bio-Rad CFX Manager (Version 3.1). The information of the primers is provided in Supplementary Table 5.

### CUT&Tag library preparation and CUT&Tag-qPCR
CUT&Tag was performed as according to Henikoff's protocol[61]. Briefly, for each sample, $2 \times 10^5$ fresh cells were resuspended in wash buffer and incubated with 10 μl pre-washed ConA beads (Bangs Laboratories, Cat.#14794) at room temperature (RT) for 10 min. The beads (bound to cells) were incubated with primary antibody (1:100) in antibody buffer at 4 °C overnight. H3K36me3 (Active Motif, Cat.#61021), H3K27me3 (Cell Signaling Technology, Cat.#9733), H3K27ac (Cell Signaling Technology, Cat.#8173), H3K9me3 (Abcam, Cat.#ab8898), H3K4me3 (Active Motif, Cat.#39159), H3K4me1 (Cell Signaling Technology, Cat.#5326), H3.3 (proteintech, Cat.#13754-1-ap), H2AK119Ub (Cell Signaling Technology Cat.# 8240), SUZ12 (Cell Signaling Technology Cat.#3737), and RING1B (Cell Signaling Technology Cat.#5694) were used. Secondary antibody [rabbit-anti-mice IgG H&L (Abcam, Cat.#ab46540) or donkey-anti-rabbit IgG H&L (sigma, Cat.#SAB3700932) according to the origin of primary antibody] and pA-Tn5 (with ME-A1/B1 adapter) were incubated at RT for 1 h at a molar ratio of 1:2. The beads were washed once with Dig-wash buffer and incubated with the secondary-antibody–pA-Tn5 complex (1:200) in Dig-300

buffer at RT for 1 h. After 3 washes, tagmentation was performed at 37 °C for 1 h. After a quick wash with TAPS wash buffer (10 mM TAPS, pH 8.5; 0.4 mM EDTA), reactions were stopped by adding Dig-300 buffer containing 15 mM EDTA, 0.1% SDS, and 100 µg/ml proteinase K and and incubating at 50 °C for 1 h with shaking. DNA was extracted with the DNA Clean & Concentrator-5 Kit (ZYMO, Cat.#D4013), and library PCR amplification was performed with NEBNext 2× PCR Master Mix (NEB, Cat.#M0541S) and standard Illumina Nextera indexing primers. The libraries were purified with Ampure XP beads at a ratio of 1:1 and sequenced using PE150 on the Illumina NovaSeq 6000. CUT&Tag-qPCR was performed as Chen reported[31]. Briefly, the DNA products of CUT&Tag were divided, one half was used as "Input", and the other undergoing library PCR amplification and purification was used as "IP products." The primers for specific regions are provided in Supplementary Table 6.

## Chromatin immunoprecipitation–deep sequencing (ChIP-seq)
ChIP-seq was performed according to Zhang's protocol[62]. Cultured cells were crosslinked with formaldehyde (final concentration 1%) for 10 min and quenched with glycine (125 mM) for 5 min at RT. The cells were scraped and collected by centrifugation, then washed once with PBS. The pellets were resuspended in cell lysis buffer and incubated on ice for 10 min. Cells were centrifuged, washed with MNase digestion buffer, and resuspended in MNase digestion buffer containing proteinase inhibitor cocktails (Roche, Cat.#4693132001). The lysates were incubated with MNase (NEB, Cat.#M0247S) at 37 °C for 20 min with continuous shaking in a thermal mixer (Fisher Scientific). The same volume of sonication buffer was added to stop the digestion. The lysates were sonicated for 3 min (10 s on/10 s off) using a Scientz-IID at 60 w and centrifuged at 20,000 × $g$ for 10 min at 4 °C. The supernatants were collected, and the chromatin content was estimated via Qubit assay (Invitrogen, Cat.#Q33231). Then 2 µg anti-EZH2 antibody (Cell Signaling Technology, Cat.#5246) was added to the chromatin in dilution of 1:100 and incubated on a rocker overnight at 4 °C. Protein A/G-magnetic beads (30 µl MCE, Cat.#HY-K0202) were added for 3 h incubation. The beads were washed with the following buffer twice each, ChIP buffer, high salt buffer, LiCl buffer, and TE buffer. Bound chromatin was eluted into elution buffer at 65 °C for 15 min. DNA was treated with RNase A and proteinase K to reverse cross-links and then purified using the DNA Clean & Concentrator™-5 (ZYMO) kit. ChIP-seq libraries were prepared from 10 ng EZH2 ChIP and input DNA using the KAPA Hyper prep kit (Roche, Cat.#KK8504). ChIP-seq libraries were sequenced to 150 bp from both ends on an Illumina NovaSeq 6000.

## Enrichment and sequencing of protein-associated nascent DNA (eSPAN) and BrdU-IP-ssSeq
eSPAN and BrdU-IP-ssSeq was performed in MCF-7, HEK293T and T47D cells according to Zhang's protocol[7,27]. Briefly, exponentially growing cells were treated with BrdU (sigma, Cat.#B5002) at 50 µM for 1 h and harvested. $1 \times 10^6$ cells were taken for eSPAN and BrdU-IP-ssSeq, respectively. For eSPAN, cells were incubated with 30 µl pre-washed ConA beads at RT for 20 min. Next, the beads were incubated with 2 µl primary antibody (H3K36me3, Active Motif, Cat.#61021) in 200 µl antibody buffer at 4 °C overnight. Secondary antibody [rabbit-anti-mice IgG H&L (Abcam, Cat.#ab46540)] and pA-Tn5 (with ME-A1 adapter) were incubated at RT for 1 h at a molar ratio of 1:2. The beads were washed once with Dig-wash buffer and incubated with 2 µg secondary-antibody–pA-Tn5 complex in 200 µl Dig-300 buffer at RT for 1 h. After 3 washes with Dig-300 buffer, tagmentation was performed at 37 °C for 1 h. After a quick wash with TAPS wash buffer, reactions were stopped by adding Dig-300 buffer containing 15 mM EDTA, 0.1% SDS, and 100 µg/ml proteinase K and and incubating at 37 °C overnight with shaking. For BrdU-IP-ssSeq, the genomic DNA was purified with phenol-chloroform-isoamyl alcohol and dissolved in ddH₂O containing RNase A (40 µg/ml). After 15 min incubation at 37 °C, 500 ng

genomic DNA was taken for in vitro tagmentation reaction [200 µl 1 × TB buffer (10% *N, N*-dimethylformamide; 10 mM TAPS•NaOH, pH 8.5; 5 mM MgCl₂) with 2 µl pA-Tn5-ME-A1 adapter complex (5.5 µM)]. After incubation at 37 °C for 30 min with gentle shaking, the reactions were stopped by adding 15 mM EDTA, 0.1% SDS, and 100 µg/ml proteinase K and incubating at 37 °C overnight with shaking. For both eSPAN and BrdU-IP-ssSeq, DNA was extracted with phenol-chloroform-isoamyl alcohol and dissolved in 32 µl ddH₂O. Oligos ME-B1 were replaced onto CUT&Tag and DNA-fragment products using an annealing program (50 °C, 1 min; 45 °C, 10 min; ramp to 37 °C at 0.1 °C/s and hold), and the gap was repaired with T4 DNA polymerase and ampligase at 37 °C for 1 h. Five percent of each sample was taken for CUT&Tag analysis, the rest was boiled for 5 min and immediately chilled on ice for 5 min. Samples were mixed with 1 ml icecold BrdU IP buffer, 1 µl *E. coli* tRNA (sigma, 10 mg/ml), and 1.5 µl BrdU antibodies (BD, Cat.#555627) and rotated for 2 h at 4 °C. Next, we added 30 µl prewashed protein G beads (GE Healthcare, 17061802) to each sample and rotated for 1 h at 4 °C. The beads were washed three times with BrdU IP buffer and once with 1× TE buffer and then incubated with elution buffer (1× TE, 1% SDS) at 65 °C for 15 min to elute the DNA. DNA in the supernatant was purified with the DNA Clean & Concentrator-5 Kit (ZYMO), and library PCR amplification was performed with NEBNext 2× PCR Master Mix and standard Illumina Nextera indexing primers. Libraries were purified with Ampure XP beads at a ratio of 1:1 and sequenced using PE150 on the Illumina NovaSeq 6000.

## Barcoding for cell lines and single-cell RNA sequencing (scRNA-seq)
MCM2-2A mutant and WT MCF-7 cell lines were barcoded via lentiviral infection to track the clone evolution following Klein's protocol[63]. pLARRY-EGFP was gifted from Dr. Fernando Camargo's lab (Addgene plasmid # 140025, RRID: Addgene_140025). A third-generation lentivirus system (pMD2.G, Addgene_12259 and PsPAX2, Addgene_12260) was used for lentivirus packaging. After lentivirus-infected cells (MCM2-2A mutant and WT) recovered for 72 h, GFP-positive cells were sorted out by BD FACSAria III and cultured for 7 additional days. To estimate the diversity of the barcodes, 10,000 cells were randomly drawn from each sample and cultured for 24 h. All cells were loaded onto a microfluidic chip and scRNA-seq libraries were constructed according to the manufacturer's instructions (Singleron GEXSCOPE Single Cell RNAseq Library Kit, Singleron Biotechnologies). Each library was generated with a unique sample index and sequenced with PE150 on the Illumina Novaseq 6000.

## Xenograft assays and single-cell RNA sequencing (scRNA-seq)
Five-to-six-week-old female NOD/ShiLtJGpt-*Prkdc^em26Cd52^Il2rg^em26Cd22^*/Gpt (NCG) mice were purchased from GemPharmatech Co., Ltd. All mice were housed in the specific pathogen-free (SPF) room under controlled temperature (20–26 °C) and humidity (40–70%) conditions with 12/12 h light/dark cycle. All xenograft experiments were performed in accordance with a protocol approved by the Institutional Animal Care and Use Committee (IACUC) at Shenzhen Institutes of Advanced Technology, Chinese Academy of Sciences (SIAT-IACUC-200319-HCS-TCC-A1148). Two days prior to cell implantation, estrogen pellets (0.72 mg/pellet 17β-estradiol, 60-day release, Innovative Research of America) were implanted subcutaneously on the animals' backs according to the manufacturer's instructions. For orthotopic xenograft studies, barcoded MCM2-2A mutant and WT MCF-7 cells were expanded in culture for the minimum time period needed to obtain a sufficient number of cells to set up replicate experiments. Barcoded MCF-7 cells ($5 \times 10^6$ cells in each replicate) were resuspended in 60 µl 1:1 mix of PBS and matrigel and injected into the fourth mammary fat pads of the mice. Mouse survival was closely monitored during the entire experimental period. Tumors were measured with Vernier calipers twice a week, and tumor volume was calculated as

0.52 × length × width². Tumor size was monitored using the Caliper Spectrum IVIS Imaging System (Xenogen, US, Version 4.0) and the entirety images of tumors were taken 7 weeks post-transplantation using Canon EOS REBEL T1i. After tumor cells had been implanted for 4 or 7 weeks, mice were euthanized and tumors were dissociated to single cells using an enzyme mixture [0.4 mg/ml Liberase™ (Roche, Cat.#5401135001); 0.75 mg/ml collagenase I (Solarbio, Cat.#C8140), 0.75 mg/ml collagenase II (Solarbio, Cat.#C8150), 0.75 mg/ml collagenase IV (Solarbio, Cat.#C8160)] at 37 °C with intermittent pipetting. For each sample, 20,000 cells were loaded and barcoded with the 10× Chromium Single Cell platform using the Chromium Single Cell 3' Library, Gel Bead, and Multiplex Kit (10× Genomics), according to the manufacturer's instructions. Each library was generated with a unique sample index and sequenced with PE150 on the Illumina Novaseq 6000 by BerryGenomics Co., Ltd. A group of the mice were euthanized 7 weeks post-transplantation to obtain the tumor tissue for further molecular experiment. Another group of the mice was closely monitored until the end point of the experiment (death, the tumor size larger than 1500 mm³, >20% body weight loss, or 79 days post-transplantation) to evaluate their survival time. The tumors and lungs were collected for histopathologic analysis as described previously[64]. Briefly, after fixing with 4% paraformaldehyde for 24–48 h, dehydration and paraffin embedding, the tissues were sectioned into thin pieces (6–8 μm). The sections were stained with Hematoxylin and Eosin Staining Kit (Beyotime, Cat.#C0105M) according to the manufacturer's instructions. Then the images were collected with the microscope (Nikon e200, Capture2.2).

### 3D organoid culture

MCM2-2A mutant, POLE3 KO and WT MCF-7 cells, as well as MCM2-90A and WT T47D cells were resuspended in complete medium and seeded into ultra-low attachment 96-well microplates (3000 cells/well). Three days later, the organoids formed. The number of viable cells was detected everyday by an automated microplate spectrophotometer (BioTek Synergy H1, USA) using the Cell Counting-Lite 3D Luminescent Cell Viability Assay (Vazyme, Cat.#DD1102) according to the manufacturer's instructions. To evaluate the viability of the organoids, Propidium Iodide [PI (Thermos Fisher, Cat.#P3566)] and Hoechst (Beyotime, Cat.#C1022) were added to the organoids (dilute at 1:300) post 3D culture for 8 days. Images were collected 15 min later under inverted microscope (EVOS M5000). To evaluate the cell proliferation, the organoids were treated with EdU (Beyotime, Cat.#ST067) at 50 μM for 5 h. The organoids were trypsinized into single cells and fixed with 70% ethanol in PBS at −20 °C overnight. The cells were pellet and resuspended in PBS containing 0.2% triton X-100. After 2 washes with PBS, Click-it reaction was performed according to the manufacturer's instructions (Beyotime, Cat.#C0081S). After 2 washes with PBS, the cells were treated with PI (1:500) and 20 μg/ml RNase A for 30 min at room temperature, protected from light. Samples were stored at 4 °C and protected from light until analysis by flow cytometry (BECKMAN COULTER CytoFLEX S and Cyt Expert 2.3). The gating strategy for FACS was shown in Supplementary Fig. 15.

### Data analysis

**RNA-seq analysis.** Samples were pooled and sequenced with paired-end (2 × 150 bp) sequencing on an Illumina NovaSeq 6000. All raw data were trimmed with Trimmomatic (v.0.39)[65] and aligned to the reference genome hg38 and assigned gene annotations from GENCODE.vM20 for *Homo sapiens*, using STAR (v.2.7.7a) software[66] with default settings. Read counting and differential gene expression analysis were carried out with Cufflinks (v.2.2.1)[67]. Cutoffs of $P$ value < 0.05 and absolute log₂ (fold change) >0.6 were adopted to identify differentially expressed genes. Functional enrichment analysis for differentially expressed genes was performed using the topGO (v.2.40.0) R package and MetaScape 3.5[68]. Significantly enriched GO terms

associated with biological processes were identified using the Fisher classic algorithm, and top enriched pathways were visualized. TBtools-II (v1.115) was used for heatmap analysis and Venn plot. Two independent clones (MCF-7) or two replicates (HEK293T and T47D) for each group were sequenced and analyzed.

**ATAC-seq, CUT&Tag, and ChIP-seq analysis.** In this study, ATAC-seq was used to evaluate chromatin accessibility. CUT&Tag and ChIP-seq were used to study the chromatin-binding affinity of histone variant, histone marks, and their associated complex members. In both MCM2-2A mutant and WT MCF-7 cells, we employed ATAC-seq; CUT&Tag for H3.3, H3K27me3, H3K27ac, H3K9me3, H3K4me3, H3K4me1, H3K36me3, SUZ12, H2AK119ub, and RING1B; and ChIP-seq for EZH2. All samples were pooled and sequenced with paired-end (2 × 150 bp) sequencing on an Illumina NovaSeq 6000. All raw data were trimmed with Trimmomatic (v.0.39)[65] and aligned to reference genome hg38 using bowtie2 (v.2.4.2)[69]. Further, uniquely aligned reads were extracted using Samtools (v.1.7)[70] and normalized to library size (Fragments Per Kilobase per Million mapped fragments, FPKM). H3K27me3, H3K9me3, H3K36me3, and H2AK119ub peaks were detected using SICER[71] (v.1.0) (https://github.com/dariober/SICERpy), with a false discovery rate cutoff of 0.01, window size 200 bp, and gap size 600 bp. H3K4me3, H3K4me1, H3K27ac, H3.3, EZH2, SUZ12, and RING1B peaks were detected using MACS2 (v.2.2.7.1)[72] with a $P$-value cutoff of 0.01. Bedtools (v.2.29.2)[73] was used to generate BigWig files and counting matrices in bins around peak summits, and bedGraphToBigWig (v.4)[74] was used for visualization. The DiffBind R package (v.3.0)[75] was used to identify peaks that differed between MCM2-2A mutant and WT cells, with cutoffs of $P$ value < 0.01 and absolute log₂ [fold change (MCM2-2A/WT)] > 1. Principal component analysis (PCA) was implemented in R (v.4.03) and 3D PCA plots were visualized using the rgl R package (v.0.105.13). The Homer suite (v.2.0)[76] was used to annotate peaks. Two independent clones in each group were sequenced and analyzed.

Signal quantification was implemented mainly using custom Perl (v.5.26.2) scripts. These scripts calculated normalized densities of H3K36me3 for peak regions (±10 kb from peak center) in which H3K27me3 was upregulated, stable, or downregulated in MCM2-2A mutant vs. WT cells. The scripts also calculated normalized densities of H3K36me3 for gene body regions (from TSS to TES) associated with promoters at which H3K4me3 was upregulated, stable, or downregulated in MCM2-2A mutant vs. WT cells, with a scan bin size of 100 bp. FPKM for defined regions, such as promoters and enhancers, were calculated with custom Perl scripts.

In addition, CUT&Tag for H3K27me3, H3K4me3 and H3.3 were performed in WT and MCM2-2A mutant HEK293T cells, as well as in WT and MCM2-90A mutant T47D cells. The data processing methods are the same as described above.

Topologically associated domain analysis of MCF-7 and T47D H3K27me3 was implemented using custom python scripts with published data. Hi-C processed data of MCF-7 WT cells were downloaded from GEO under accession number GSM1631185[29], genome transforming from hg19 to hg38 was performed with the UCSC liftOver tool (http://genome.ucsc.edu/cgi-bin/hgLiftOver). Similarly, Hi-C matrix data of T47D WT cells were downloaded from ENCODE (https://www.encodeproject.org) under accession number ENCSR549MGQ[77] and further processed by Juicer tools (v2.20.00)[78].

Enhancers were identified based on the positions of H3K27ac, H3K4me3, and H3K4me1 peaks. All H3K4me1 peaks depleted of H3K4me3, with an exclusion region distance to the TSS set at 2.5 kb, were considered enhancer regions; those that overlapped H3K27ac peak regions were considered active enhancers, and the rest were considered poised enhancers[79]. Poised enhancers decorated with H3K27me3 were considered repressed enhancers. Super enhancers were identified using ROSE (v1) based on the H3K27ac ranking signal[80].

For each histone mark, GO terms significantly associated with genes that differed between MCM2-2A mutant vs. WT cells were identified, as described in the RNA-seq analysis section. Two replicates for each sample were sequenced and analyzed.

The relative enrichment ratios of annotated peaks for selected histone marks in MCM2-2A vs. WT cells calculated as the formula:

$$\text{Enrichment ratio (DP, SP)} = \log_2\left(\frac{\text{obs}_d/\text{exp}_d}{\text{obs}_s/\text{exp}_s}\right) \quad (1)$$

where DP means differential peaks (upregulated or downregulated) of a selected histone mark in MCM2-2A mutant vs. WT cells, and SP means corresponding stable peaks. obs means observing total length of annotated differential or stable peaks of certain types, including promoter, 5' UTR, exon, intron, TTS, 3' UTR, enhancer, intergenic, LINE, SINE, LTR, simple repeat, satellite and CpG island. exp means the corresponding experimental length of peaks of certain types, evaluating by:

$$\exp = L_p/L_g \times L_t \quad (2)$$

where $L_p$ means experimental length of peaks, $L_g$ means length of whole genome, $L_t$ means length of target peaks. Target peaks represent the differential or stable peaks of selected histone marks.

**scRNA-seq analysis.** scRNA-seq was implemented on an Illumina NovaSeq 6000. Raw data were processed using the Kallisto-bustools workflow (v.0.46.0)[81], including the steps of associating reads with cells, collapsing reads based on unique molecular identifiers (UMIs), and generating gene counts in cells. For data quality control, genes detected in fewer than three cells and cells with fewer than 200 genes captured were removed from analysis. Furthermore, cells with high (>8000) or low (<2000) numbers of detected UMIs and cells with more than 5% mitochondrial gene expression were filtered out, to avoid multiplets, broken droplets, and the influence of apoptosis. After quality control, data was imported into the Seurat R package[82] (v.4.0.3) (https://satijalab.org/seurat/), then normalized and scaled with the NormalizeData and ScaleData functions. The FindVariableFeatures function was used to identify highly variable genes, the FindNeighbors function was used to construct a shared nearest neighbor graph, and the FindClusters function was employed to identify clusters of cells. Combined with the t-distributed stochastic neighbor embedding (t-SNE) method, the RunTSNE and DimPlot functions were used for dimension reduction and cell cluster visualization. Marker genes or genes that differed between mutant vs. WT cells were identified with the FindMarkers function, using cutoffs of P value < 0.01 and absolute Log10 (fold change) >0.3. Average gene expression of clusters or samples was calculated with the AverageExpression function. The VlnPlot, FeaturePlot, and DoHeatmap functions in Seurat (v.4.0.3) combined with the ggplot2 R package[83] (v2_3.3.5) were used to visualize the expression of selected genes across clusters and samples.

**Clone calling.** A published method, LARRY (https://github.com/AllonKleinLab/LARRY)[63], was adopted to call lineage clones (i.e., those with cells from the same ancestor). We used custom python (v.3.8) scripts to extract all cell barcodes, UMIs, and lineage barcodes from raw files for clone calling. Using the custom python script clonal_annotation.py provided by the LARRY pipeline, we retained cell barcode–UMI–lineage barcode triples supported by at least 10 reads, lineage barcodes with a hamming distance of 3, and at least 3 UMIs.

General statistics for clone size were calculated separately for MCM2-2A and WT cells after lineage clones were called. Clone size ranged from 2 to 949 cells in MCM2-2A mutant tumors and 2–251 cells in WT tumors. Specifically, clones were divided into three groups of cell size 2–10, 11–20, and >20. Clones in the >20 group were considered as dominant clones for further analysis. Dominant clone marker genes are identified by comparing the expression profile of MCM2-2A mutant dominant clones vs. all WT tumor cells at both 4 and 7 weeks.

**Gene set variation analysis (GSVA).** Enrichment pathway analysis was implemented in the GSVA R package (v1.38.2)[84,85] with the default settings based on scRNA-seq data. Cancer hallmark and GO biological pathway datasets were obtained from the Molecular Signatures Database (http://www.gsea-msigdb.org/gsea/msigdb)[86] and exported using the msigdbr R package (v7.4.1) (https://igordot.github.io/msigdbr/). Differences in pathway activity were scored in each cell of clusters.

Cancer functional state score was calculated based on the data sets from CancerSEA (http://biocc.hrbmu.edu.cn/CancerSEA/)[36]. Breast cancer specific gene sets summarized by published scRNA-seq projects of 14 cancer functional states, including angiogenesis, apoptosis, cell cycle, differentiation, DNA damage, DNA repair, EMT, hypoxia, inflammation, invasion, metastasis, proliferation, quiescence and stemness were downloaded and calculated in a GSVA method as described above.

**Breast cancer subtype identification.** A 50-gene subtype predictor (PAM50) was applied to scRNA-seq clinical subtype identification. Each single cell was identified as one of the basal-like (Basal), HER2-enriched (HER2), luminal A (LumA), luminal B (LumB), or undefined type. The classification was implemented by the supervised risk PAM50 predictor[37]. Detailed processing methods are based on pam50_centroids.txt and R script predict.R provided in the following link: https://static-content.springer.com/esm/art%3A10.1186%2Fs12864-019-5849-0/MediaObjects/12864_2019_5849_MOESM6_ESM.zip.

**eSPAN analysis.** The bias resolving rate of H3K36me3 in MCM2-2A mutant and WT cells was measured with eSPAN[7] and calculated for human core origins ($n = 7624$) identified previously[87]. Partition or upward (downward) slopes in replication fork directionality were calculated with the cal_bias.R and draw_bias.R scripts provided by the eSPAN pipeline (https://github.com/clouds-drift/eSPAN-bias)[27]. The bias of H3K36me3 eSPAN at selected replication origins was computed from unique reads in each bin using the formula

$$\text{Bias} = (W - C)/(W + C) \quad (3)$$

where $W$ and $C$ are the number of reads mapped onto the Watson and Crick strands in each bin, respectively. Bias was calculated with a default bin size of 100 bp, normalized by BrdU signal, and further smoothed with the 1000 bins flanking each side for visualization.

## Statistics and reproducibility

CUT&Tag for H3K27me3, H3K4me3, H3K36me3, H3K9me3 H3K27ac, H3K4me1, H3.3, H2AK119Ub, SUZ12, and RING1B, ATAC-seq, ChIP-seq and bulk RNA-Seq were performed in two independent clones of WT and MCM2 mutant MCF-7 cells. Besides, CUT&Tag for H3K27me3 in WT and MCM2 mutant MCF-7 cells was performed twice independently. CUT&Tag for H3K27me3 and H3K4me3, as well as bulk RNA-Seq was performed in two replicates for WT and MCM2 mutant HEK293T and T47D cells. CUT&Tag for H3K27me3 was performed in two replicates for POLE3 KO MCF-7 cells. PCA analysis indicated the results were consistent and reproducible between CUT&Tag or bulk RNA-Seq replicates. Western blot was repeated three times over two independent clones in WT and MCM2-2A mutant MCF-7 cells and twice in WT and MCM2 mutant HEK293T and T47D cell lines, as well as in POLE3 KO MCF-7 cells. To measure the size of tumors and analyze the survival time of mice bearing tumors, xenograft assay was performed twice independently. qPCR experiments were performed in three

replicates or more independently as described in the associated figure legends. 3D organoid culture was performed in four independent experiments in each genotype in MCF-7 and T47D cell lines. Besides, 3D organoid culture for growth curve and organoids' PI/Hoechst staining assays were repeated twice in WT and MCM2-2A MCF-7 cells. All the replicated experiments were performed independently and obtained similar results. The detailed statistical test for each experiment was displayed in the associated figure legends.

## Reporting summary

Further information on research design is available in the Nature Portfolio Reporting Summary linked to this article.

## Data availability

The data that support this study are available from the corresponding author upon reasonable request. The deep sequencing data generated in this study have been deposited in the Gene Expression Omnibus (GEO) database under accession code GSE201262, including all the raw data and processed data. The Hi-C processed data of WT MCF-7 and T47D cells used in this study are available in the GEO database under accession code GSM1631185 and ENCODE database under accession code ENCSR549MGQ, respectively (detailed in the "Methods" section). The gene sets related to breast cancer invasion, metastasis and progression in patients was obtained from Molecular Signatures Database (https://www.gsea-msigdb.org/gsea/msigdb/genesets.jsp?collection=CGP). Cancer hallmark datasets were obtained from Molecular Signatures Database (https://www.gsea-msigdb.org/gsea/msigdb/human/genesets.jsp?collection=H). Cancer functional state score of breast cancer single cells was calculated based on the data sets from CancerSEA (http://biocc.hrbmu.edu.cn/CancerSEA/). Large source data sheets were deposited in Zenodo database under accession code 7927636. Source data are provided with this paper.

## Code availability

We have made use of publicly available software and tools. The published code of normalization and PAM50 scripts are available with the following link: https://static-content.springer.com/esm/art%3A10.1186%2Fs12864-019-5849-0/MediaObjects/12864_2019_5849_MOESM6_ESM.zip. The published pipeline used to calculate bias via eSPAN analysis is available with the following link: https://github.com/clouds-drift/eSPAN-bias. The published scripts used to call lineage barcodes are available with the following link: https://github.com/AllonKleinLab/LARRY. All other codes used to generate the analysis have been placed in Zenodo through the following link: https://zenodo.org/record/7927636.

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

## Acknowledgements

This work was supported by the following fundings: National Key R&D Program of China (Grant No. 2019YFA0903800 to H.G.), the Major Program of the National Natural Science Foundation of China (32090031 to H.G.), the Strategic Priority Research Program of the Chinese Academy of Sciences (XDB0480000 to H.G.), the General Program of the National Natural Science Foundation of China (32070610 to H.G.), the National Natural Science Foundation of China for Young Scholars (32000580 to Q.W., 32100460 to J.Z. and 32101178 to Y.Y.), the Guangdong Province Fund for Distinguished Young Scholars (2021B1515020109 to H.G.), Guangdong Basic and Applied Basic Research Foundation (2021A1515110377 to X.K.), Shenzhen Institute of Synthetic Biology Scientific Research Program (JCHZ20200005, DWKF20210001, ZTXM20190019 to H.G.), and Project funded by China Postdoctoral Science Foundation (2021M693301 to C.T and 2021M703386 to J.Z.).

## Author contributions

H.G., C.T., J.Z., and Y.G. conceived the project. C.T., X. Li, X.K., N.W., G.S., S.H., J.J., and J.L. performed experiments. Q.W., Y.Y., C.Y., Z.W., X. Liu, X.P., K.C., and Z.L. helped with the design of experiments. J.Z. performed the data analysis. H.G. supervised the study. C.T., Y.G., and T.Z. wrote the manuscript with comments from all authors.

## Competing interests

The authors declare no competing interests.
