## [Peer Review File · Nature Communications]

Impaired histone inheritance promotes tumor progressionREVIEWER COMMENTS

Reviewer #1 (Remarks to the Author):

In this manuscript by Tian et al, the authors constructed an impaired histone inheritance model by mutating MCM2-2A. Using this model, they show that epigenetic patterns are reprogrammed, which also results in transcriptional rewiring. This altered program is suggested to provide cells a growth advantage. Overall, the questions seem interesting and employ several high throughput methods. However the experiments are performed in a single model system (MCF7) with limited replicates, making it difficult to perform any meaningful statistical analysis. Whilst i understand the time and effort involved in performing experiments and analyzing them, i feel that it is necessary to repeat at least key experiments (such as histone profiling and RNA-seq) in an additional model system with appropriate numbers of replicates if possible.

Some additional points highlighted below.

Figure 2c - The RNA-seq results look to be in the right direction, but i don't see any statistical analysis. I would want to see fold change statistics between KO and WT cells for each group.

Fig3a - Highly enriched peaks showed no differences between KO and WT, while weaker peak regions changed. This could be biologically significant as suggested by authors, but also likely influenced by technical differences which are usually amplified in weaker peaks. Thus, without an additional cell line or robust statistical tests from multiple replicates, i don't think this claim is valid.

The mouse experiments seem well done, with appropriate replicates and interpretation.

The single-cell data clusters poorly and substantial claims between clusters cannot be made. Minor adjustments to clustering parameters will give caring numbers of clusters in cases like this where the groups are not distinct enough. I would thus recommend simplifying the single-cell analysis and caution over-interpretation. S6F for example does not show strong enrichments.

Finally, to add significance to these results, it would be interesting to connect them to any existing patient data that suggests epigenetic reprogramming or clonal evolution in patient samples. Does increased mutations in these proteins provide fitness advantage in published single-cell DNA-seq data? Do some of these clusters and results align with published scRNA-seq data in patients, especially those with metastasis? Patten et al Nature medicine (2018) have published enhancer regions in breast cancer patients and show regions that are clonal vs sub clonal. Would these region correspond to regions changing in these experiments? I am not suggesting the authors perform all these analysis, but any patient relevant data would be highly informative.

Reviewer #2 (Remarks to the Author):

In this article, the authors constructed a MCM2-2A mutation in MCF-7 breast cancer cells and examined the impaired histone inheritance. They found the repressive histone mark H3K27me3 was notably reprogrammed, and the correlated gene expression was altered. Finally, they transplanted the mutant cells to mice and observed abnormal tumor growth and metastasis. However, based on the presented data, it is not rigorous to directly make a conclusion that the impaired histones inheritance promotes tumor growth and metastasis

1. To verify the accuracy of data that obtained from RNA-seq and ChIP-seq, RT-qPCR and ChIP-qPCR should be considered to perform. Especially for the selected overlapped genes.

2. To prove the impaired parental histone inheritance in MCM2-2A cell lead to abnormal tumor growth and metastasis, or clone formation, related cellular functional experiments should be considered to carry out. Also, involved gene expression should be confirmed by RT-qPCR and WB.

3. The authors did not describe the statistic method of Kaplan-Meier survival analysis, while the curves were crossed over, and the log-rank test may not be appropriate (Figure 4e).

Minor errors:

The grammars and tenses should be carefully checked.

Eg: Line 154/178, they should be past tense.

Eg: Line 142, redundant "both".

Eg: Line 197-198, the description of "The expression level of genes with H3K4me3 stable promoters was also higher in both MCM2-2A mutant and WT cells" is not accurate.

The part of introduction should be described in a more concise way.

Reviewer #3 (Remarks to the Author):

In this paper, Tian et al describes how impairment of histone inheritance influences gene expression and tumor progression using MCF-7 breast cancer cell line. Authors used MCM2 mutant cell line in which MCM gene's histone chaperone function is specifically disturbed (without disturbing helicase function) and analyzed gene expression changes in detail and its effect on clonal evolution. Given the fact that the gene MCM2 is involved in cell proliferation in various cancers, this study provides an important information as to how MCM2 plays a role in cancer progression. This study provides several other interesting points regarding the hierarchy of gene regulation among different histone modifications and other chromatin states (open/closed, A/B compartments). The authors used new and suitable techniques to analyze genome-wide changes of these states.

Overall, the findings should be of great interest to the readers of Nature Communications. There are some concerns about data interpretation and presentation and lack of some details of analysis. I hope my comments help the authors to make this paper more compelling.

Major points;

I suggest that each result could be provided more thoroughly without fitting to certain idea first, then give interpretations later in discussion. It is especially interesting that authors observed that majority of H3K27me3 peaks remained stable in MCM2-2A mutant cells throughout multiple cell divisions, and downregulation of H3K27me3 mainly occurs at the A-compartment. From this point of view, I have an impression that histone inheritance may not be a major factor for maintaining gene expression memory. Further interpretation and discussion will be valuable for this point.

MCM-2 mutant cells seem to have "lost" template H3K27me3 from actively transcribed regions, and show quite fixed/stable phenotype. These cells do not seem to "keep forgetting" of their memory further. In that case, what explain the mechanism of tumor evolution? MCM-2 mutant may have more phenotypic (gene expression level) or epigenetic variability than WT MCF7 cells? I believe that single cell data can be analyzed to consider the mechanism deeper.

Can authors provide more logic of each analysis and how to read graphs more for general audiences. For example, how eSPAN graph is made, what the "bias" is, would be helpful if they can spell out more clearly in the text or figure legends.

It is a good design to use MCM2 tyrosine mutants to preserve helicase function. It will be informative if they can show MCM2 mutation in homozygous for their lines (sanger sequence data) and show

helicase function (cell division?) is not perturbed in those obtained clones.

It should be informative if authors can provide any data or information why they used H3K36me3 to detect preexisting histones in their cells if possible.

Based on the method section, eSPAN data was normalized by BrdU signal. Does the "signal" mean intensity or BrdU-IP ssDNA sequencing data? Either way, they need to provide data and details.

It would be informative how many origins out of selected replication origins are active (or inactive) in their lines based on their brdu seq data and if so, is there any correlation between gene activity change relative to the distance from the origin?

MCM2 mutant lines show increased H3K27me3 signal in western blotting. If that is the case, did they normalize the H3K27me3 cut and tag data with the protein amount?

Their definition of "downregulated" "upregulated" and "stable" loci is confusing in the texts. Please give the definition more clearly for each place.

Point-by-point response to the Reviewer's comments

We thank the Reviewers for their feedback, suggestions, and critical remarks on our manuscript entitled "Impaired histone inheritance promotes tumor progression". We have revised and improved our manuscript based on their comments and believe to have addressed their suggestions and concerns. Below, please find our detailed point-by-point response to the specific comments of the Reviewers in blue and in red within the revised manuscript file. To avoid confusion, we used New Fig. # or New Supplementary Fig. # to refer to Figures in the revised manuscript and Response Fig. # to refer to Figures in this response letter.

Reviewer #1 (Remarks to the Author):

In this manuscript by Tian et al, the authors constructed an impaired histone inheritance model by mutating MCM2-2A. Using this model, they show that epigenetic patterns are reprogrammed, which also results in transcriptional rewiring. This altered program is suggested to provide cells a growth advantage. Overall, the questions seem interesting and employ several high throughput methods. However the experiments are performed in a single model system (MCF7) with limited replicates, making it difficult to perform any meaningful statistical analysis. Whilst i understand the time and effort involved in performing experiments and analyzing them, i feel that it is necessary to repeat at least key experiments (such as histone profiling and RNA-seq) in an additional model system with appropriate numbers of replicates if possible.

Response: We appreciate the reviewer for the thoughtful critiques. As suggested, we constructed two additional impaired histone inheritance models: the MCM2-2A (Y81A, Y90A) mutant HEK293T and MCM2-90A (Y90A) T47D cells. We profiled the total level and chromatin occupancy of histone marks H3K27me3, H3K4me3 and histone variant H3.3, gene expression changes, and repeated all the data analysis in these two cell lines. All the experiments were performed in two replicates. In addition, the growth of MCM2 mutant and POLE3 KO tumor organoids was monitored. We observed similar epigenetic reprogramming, differential gene expressions and accelerated growth phenotypes in additional impaired histone inheritance models as in the MCM2-2A mutant MCF-7 cells. Detailed below:

Impaired histone inheritance models construction

We obtained an MCM2-2A mutant HEK293T cell line and an MCM2-90A T47D cell line (**Response Fig. 1a, b or New Supplementary Fig. 1e, f**) and characterized the pattern of parental histone segregation using eSPAN in these cell lines. The leading-strand bias were also exacerbated in both MCM2 mutant cell lines (**Response Fig. 1c, d or New Supplementary Fig. 1g, h**) as seen in MCF7 cells, suggesting an impaired parental histone transferring to the lagging strands in both cell lines. Similar to the MCF7 cells, we also observed an increased level of H3K27me3 in these two models (**Response Fig. 1e or New Supplementary Fig. 1i**).

Response Figure 1

Response Figure 1. Impaired parental histone inheritance in MCM2-2A mutant HEK293T cells and MCM2-90A T47D cells.

a, b Sanger sequencing confirming the homozygous MCM2-2A HEK293T colony (**a**) and homozygous MCM2-90A T47D colony (**b**). **c** Average strand bias of H3K36me3 in MCM2-2A mutant and WT HEK293T cells, as determined using eSPAN. **d** Average strand bias of H3K36me3 in MCM2-90A and WT T47D cells determined by eSPAN. **e** Western blot analysis showing select histone marks and histone variant H3.3 in MCM2-2A mutant and WT HEK293T cells, as well as in MCM2-90A and WT T47D cells.

Disturbing histone inheritance reprograms H3K27me3 pattern

We first compared the changes in H3K27me3 peaks among the three MCM2 mutant models we generated in MCF-7, HEK293T and T47D cells, and found very similar pattern in the downregulated H3K27me3 regions (**Response Fig. 2a** or **New Supplementary Fig. 6a**). However, we observed highly similar H3K27me3 upregulated regions between MCF-7 and T47D cell lines, but not HEK293T (**Response Fig. 2b** or **New Supplementary Fig. 6b**), suggesting a more complicated

and undefined mechanism for H3K27me3 upregulation. Then, we annotated the H3K27me3 peaks in T47D cells based on their genomic locations and found that the majority of upregulated H3K27me3 peaks were also distributed in B-type compartments, whereas downregulated H3K27me3 peaks were mainly distributed in A-type compartments, similar to that seen in MCF7 cells (**Response Fig. 2c or New Supplementary Fig. 6c**). Since the published Hi-C data for distinguishing compartment A and B in HEK293T cells was not available, we could not perform this analysis in HEK293T cells. Next, we explored the regulatory function of altered H3K27me3 on its target gene in these two cell lines and found that the expression level of genes with H3K27me3 downregulated promoters increased significantly, while the expression of genes with upregulated H3K27me3 remained unchanged (**Response Fig. 2d, e or New Supplementary Fig. 8b**).

Response Figure 2

Response Figure 2. The similarity of H3K27me3 reprogramming among MCM2 mutant line cells.

a, b Boxplot showing the H3K27me3 signal in indicated cells at H3K27me3 downregulated (**a**) or upregulated peaks (**b**) in MCM2-2A mutant MCF-7 cells. #1 and #2 indicate two independent clones, R1 and R2 indicate two replicates. **c** Proportion of H3K27me3 peaks belonging to A-type vs. B-type compartments, stratified by whether the peaks were upregulated, stable, or downregulated in

MCM2-90A T47D cells. **d, e** Heatmap showing signal intensity of H3K27me3 at promoters (*left*) and the expression of their target genes (*right*) in MCM2-2A mutant and WT HEK293T cells (**d**), as well as in MCM2-90A and WT T47D cells (**e**).

Transcriptionally active chromatin remains stable despite impaired parental histone inheritance.

We investigated the original epigenetic profiles of the H3K4me3 upregulated, stable, and downregulated peaks in MCM2 mutant cells and found that the intensity of H3K4me3 at the stable peaks was significantly higher than those at up- or downregulated peaks in both HEK293T and T47D cell lines (**Response Fig. 3a, b or New Supplementary Fig. 9a, b**). Consistently, these H3K4me3 stable regions had faster histone turnover marked by higher H3.3 levels (**Response Fig. 3c, d or New Supplementary Fig. 9d, e**). The expression levels of genes with H3K4me3 stable promoters were also higher than those with H3K4me3 up- or downregulated promoters in both HEK293T and T47D cell lines (**Response Fig. 3e, f or New Supplementary Fig. 9g, h**). Similarly, transcriptionally active regions remain stable in the two additional models we generated.

Response Figure 3

Response Figure 3. H3K4me3 remains stable at more active chromatin regions in MCM2 mutant HEK293T and T47D cell lines.

a, b H3K4me3 signal in WT HEK293T cells (**a**) and WT T47D cells (**b**), at H3K4me3 peaks that were upregulated, stable, or downregulated in MCM2-2A mutant HEK293T cells and MCM2-90A T47D cells, respectively. **c, d** Boxplots showing H3.3 levels in WT HEK293T cells (**c**) and WT T47D cells (**d**) at peaks where H3K4me3 were upregulated, stable, or downregulated in MCM2-2A mutant HEK293T cells and MCM2-90A T47D cells, respectively. **e, f** Boxplots representing expression level of genes with H3K4me3 upregulated, stable, or downregulated promoters in MCM2-2A mutant (*right*) and WT (*left*) HEK293T cells (**e**), as well as in MCM2-90A (*right*) and WT (*left*) T47D cells (**f**). *** $P < 0.001$, ns $P > 0.05$.

Impaired histone inheritance leads to similar changes in gene expression among MCM2 mutant cell lines

We then compared the transcriptome profile changes among these three MCM2 mutant models relative to their WT counterparts and observed a significant overlap in up- or downregulated genes between the cell lines (**Response Fig. 4a, b** or **New Supplementary Fig. 7a, b**). The overlapped dysregulated genes played a vital role in gland development, epithelial cell proliferation and regulation of growth, which contributes to tumor progression (**Response Fig. 4c, d** or **New Supplementary Fig. 7c, d**).

Response Figure 4

Response Figure 4. The similarity of gene expression changes among MCM2 mutant cell lines.

a Venn plot showing the upregulated (*left*) or downregulated (*right*) genes in both MCM2-2A MCF-7 and MCM2-2A HEK293T cells compared to their WT counterparts. **b** Venn plot showing the upregulated (*left*) or downregulated (*right*) genes in both MCM2-2A MCF-7 and MCM2-90A T47D cells compared to their WT counterparts. **c, d** GO terms enriched for the dysregulated genes in both MCM2-2A MCF-7 and MCM2-2A HEK293T cells (**c**), as well as in both MCM2-2A MCF-7 and MCM2-90A T47D cells (**d**).

Impaired histone inheritance promotes tumor organoids growth

We cultured the tumor cells in organoids to mimic the *in vivo* environment and found that the MCM2-2A and POLE3 KO MCF-7 organoids grew significantly faster than WT (**Response Fig. 5a**

or New Supplementary Fig. 11a). Similarly, the MCM2-90A T47D organoids also acquired growth advantages compared to WT (Response Fig. 5b or New Supplementary Fig. 11b). The percentage of proliferating cells was higher in both MCM2-2A mutant and POLE3 KO MCF-7 organoids than WT (Response Fig. 5c, d or New Supplementary Fig. 11c, d). We observed similar changes in gene expression between POLE3 KO and MCM2-2A mutant MCF-7 organoids (Response Fig. 5e-g or New Supplementary Fig. 11e-g), and the changed genes were mainly involved in cell growth, proliferation, migration, mammary gland development and apoptotic process (Response Fig. 5f or New Supplementary Fig. 11f), which is consistent with the accelerated growth phenotypes.

Response Figure 5

Response Figure 5. Impaired histone inheritance promotes the tumor growth in MCF-7 and T47D organoids.

a The growth curve of the MCM2-2A mutant, POLE3 KO and WT MCF-7 organoids. Error bars, SD (n = 4). *** $P < 0.001$, * $P < 0.05$. One-way ANOVA test. **b** The growth curve of the MCM2-90A and WT T47D organoids. Error bars, SD (n = 4). * $P < 0.05$, Student's t test. **c** Images showing

the morphology and cell alive state of MCM2-2A mutant, POLE3 KO and WT MCF-7 organoids. Propidium iodide, PI (+) represents dead cells, Hoechst (+) represents alive cells. **d** Cell proliferation evaluated by EdU staining in MCM2-2A mutant, POLE3 KO and WT MCF-7 organoids. **e** Heatmap showing the differentially expressed genes in MCM2-2A mutant vs. WT MCF-7 organoids and their corresponding expression in POLE3 KO and WT MCF-7 organoids. #1 and #2 indicate two independent clones. R1 and R2 indicate two replicates. **f** GO enrichment for the dysregulated genes in both MCM2-2A mutant and POLE3 KO MCF-7 organoids. **g** RT-qPCR showing the expression of proliferation-related genes in MCM2-2A mutant, POLE3 KO and WT MCF-7 organoids. The genes' upregulation (*left*) or downregulation (*right*) could promote cancer cell proliferation. Error bars, SD (WT and POLE3 KO, n = 4; MCM2-2A, n = 8). * $P < 0.05$, ** $P < 0.01$, *** $P < 0.001$; Student's t test.

In summary, we profiled the epigenetic reprogramming of histone marks (H3K27me3 and H3K4me3) and transcriptional rewiring in two additional models and observed highly consistent phenotypes as our initial MCF7 model.

Some additional points highlighted below.

1. Figure 2c - The RNA-seq results look to be in the right direction, but i don't see any statistical analysis. I would want to see fold change statistics between KO and WT cells for each group.

Response: We appreciate the reviewer for raising this point. We added a heatmap showing the fold change of corresponding gene expression to **New Fig. 2c and Response Fig. 6a**. The statistics of the expression for genes with H3K27me3 up- or downregulated promoters was shown in **Response Fig. 6b or New Supplementary Fig. 8a**.

Response Figure 6

Response Figure 6. Statistic for the expression changes of genes targeted by dysregulated H3K27me3.

a Heatmap showing signal intensity differences in H3K27me3 at promoters (*left*) and corresponding changes of gene expression (*right*) in MCM2-2A mutant vs. WT MCF-7 cells. FC, fold change. **b** Boxplots representing the expression levels of genes with promoters at which H3K27me3 was upregulated, stable, or downregulated in MCM2-2A mutant vs. WT MCF-7 cells. One-way ANOVA indicates that the expression of genes with H3K27me3 downregulated promoters increased

significantly in MCM2-2A cells, whereas, those with H3K27me3 stable or upregulated promoters are low in both MCM2-2A and WT MCF-7 cells. #1 and #2 indicate two independent clones. *** represents $P < 0.001$, ** represents $P < 0.01$ and ns represents $P > 0.05$.

2. Fig3a - Highly enriched peaks showed no differences between KO and WT, while weaker peak regions changed. This could be biologically significant as suggested by authors, but also likely influenced by technical differences which are usually amplified in weaker peaks. Thus, without an additional cell line or robust statistical tests from multiple replicates, i don't think this claim is valid.

Response: We appreciate the reviewer for the suggestions. In our analysis, we used a very strict criteria ($-\log_{10} P\text{-value} \geq 50$, peak length ≥ 500) when we identified solid H3K4me3 peaks, to remove weaker regions. The high quality of the peaks minimized the technical differences in our subsequent analysis. We performed H3K4me3 CUT&Tag in 2 independent WT clones of MCF-7 cells and showed them separately in our revised manuscript (**Response Fig. 7 or New Fig. 3a**). The data of these two replicates was consistent, indicating the high reproducibility of our results. Besides, we have repeated the data analysis of H3K4me3 CUT&Tag and RNA-seq in two additional cell lines (HEK293T MCM2-2A and T47D MCM2-90A), and observed highly similar changed patterns (**Response Fig. 3 or New Supplementary Fig. 9**).

Response Figure 7

Response Figure 7. H3K4me3 signal in WT MCF-7 cells, at H3K4me3 peaks that were upregulated, stable, or downregulated in MCM2-2A mutant vs. WT cells. #1 and #2 indicate two independent WT MCF-7 clones.

3. The mouse experiments seem well done, with appropriate replicates and interpretation.

Response: We thank the reviewer's positive comments.

4. The single-cell data clusters poorly and substantial claims between clusters cannot be made. Minor adjustments to clustering parameters will give caring numbers of clusters in cases like this where the groups are not distinct enough. I would thus recommend simplifying the single-cell analysis and caution over-interpretation. S6F for example does not show strong enrichments.

Response: We appreciate the reviewer for this suggestion. We provided the data analysis process in

Response Fig. 8 to show that the quality of our scRNA-seq data is satisfactory. The detailed quality control steps were described in Methods.

In our experiment, the dissociated tumor tissue containing both human MCF-7 cells and mouse mammary gland cells were sequenced at the same time. We then divided human tumor cells and mouse cells based on the species information of their RNA sequences, and analyzed them with the same parameters (**Response Fig. 9a, b**). While mouse cells clustered well (**Response Fig. 9b**), the tumor cells clustered poorly which might be the consequence of the high similarity of the cells that derived from cell line. In support of this, we searched through the literature and analyzed the published scRNA-seq datasets for MCF-7. We found that the published MCF-7 scRNA-seq had no obvious boundaries (**Response Fig. 9c-f**), which is similar to ours in this study.

We next reanalyzed the scRNA-seq data regardless of the cluster information, and identified the differentially expressed genes at 4 and 7 weeks. We found that they are mainly enriched in the pathways related to regulation of cell growth, mitotic cell cycle, DNA damage and epithelial cell differentiation (**Response Fig. 10**), in which the marker genes of cluster 2 and 5 are enriched (**New Fig. 5d**). These results imply that the difference between MCM2-2A and WT tumor cells is mainly contributed by the cells in clusters 2 and 5. Here, analysis either with all the cells together and or after clustering draws similar conclusions, suggesting that the clustering analysis we used is reliable and our conclusion is valid.

Response Figure 8

Response Figure 8. scRNA-seq analysis processes.

a Knee plots showing the parameters for cell filter in each sample. **b, c** Violin plots showing the total counts of RNA (*left*) and the number of Feature RNA (*right*) in each sample. Cells were filtered based on the number of its Feature RNA (2000-8000).

Response Figure 9

Response Figure 9. Clustering analysis of MCF-7 and mouse cells.

a, b *t*-SNE plot showing the clusters of tumor cells (**a**) and mouse cells (**b**). The same parameters were used to reduce the dimensionality (dimension = 10) and cluster the cells (resolution = 0.15)

for tumor cells and mouse cells. **c** scRNA-seq of mammospheres (MS) derived from MCF-7 cells¹. **d** Bi-dimensional representation of single-cell transcriptomes (MCF7 and LTED)². **e** scRNA-seq of untreated (*left*) and 4OHT-treated (*right*) MCF-7 cells³. **f** scRNA-seq of breast cancer and noncancerous cell lines⁴. The orange box indicated the MCF-7 cells.

Response Figure 10

Response Figure 10. Simplified analysis of the scRNA-seq data *in vivo*.

a, b Gene ontology enrichment analysis for the differentially expressed genes in MCM2-2A mutant vs. WT tumor cells identified 4 weeks (**a**) and 7 weeks (**b**) post-transplantation. The transcriptome data of all the cells in MCM2-2A mutant and WT tumor was used to calculate the differentially expressed genes.

S6F for example does not show strong enrichments.

The reviewer mentioned that the genes in S6F does not show strong enrichments. We showed the GSVA score of each cell in each cluster in a box plot, and analyzed the significance of differences by Kruskal-Wallis test (**Response Fig. 11a**). The GSVA score of cells in cluster 2 was significantly higher than that in any other clusters ($P < 0.0001$). The color scheme we used before might lead to a poor visualization, thus, we modified the color scheme for a better display (**Response Fig. 11b** or **New Supplementary Fig. 12f**).

Response Figure 11

Response Figure 11. GSVAscore for the GO terms in each cell in each cluster.

a Boxplot showing the GSVAscore for the gene ontology terms “epithelial cell differentiation” (*left*) and “gland development” (*right*) in each cell in each cluster. *** $P < 0.001$. Kruskal-Wallis test. **b** Heatmaps showing expression profile in each cluster for genes enriched in the gene ontology terms “epithelial cell differentiation” (*left*) and “gland development” (*right*). GSVAscores are shown below the heatmaps. Each column represents a single cell.

5. Finally, to add significance to these results, it would be interesting to connect them to any existing patient data that suggests epigenetic reprogramming or clonal evolution in patient samples. Does increased mutations in these proteins provide fitness advantage in published single-cell DNA-seq data? Do some of these clusters and results align with published scRNA-seq data in patients, especially those with metastasis? Patten et al Nature medicine (2018) have published enhancer regions in breast cancer patients and show regions that are clonal vs sub clonal. Would these region correspond to regions changing in these experiments? I am not suggesting the authors perform all these analysis, but any patient relevant data would be highly informative.

Response: We appreciate the reviewer for the thoughtful suggestions. We have connected the gene expression feature of each cluster with patient data (**New Supplementary Fig. 12g lower panel**). We obtained the dysregulated gene sets that are significantly associated with invasion, metastasis or prognosis in breast cancer patients from the GSEA website and analyzed the GSVAscores of these gene sets in each cell. Cells in cluster 5 owned a higher score, suggesting an aggressive phenotype (**New Supplementary Fig. 12g**). We emphasized this in our revised manuscript.

Then, we followed the suggestions of the reviewer to connect our data with the scRNA-seq data in patients. We characterized the breast cancer function state of our single cells based on CancerSEA⁵, which portrays human cancer single-cell functional state atlas. The cells in cluster 5 got higher scores of function states involving cell cycle, EMT, invasion and proliferation (**Response Fig. 12a**

or **New Supplementary Fig. 12h**), which further supports the functional importance of this cluster in the tumor progression.

To evaluate the recurring gene expression features, we classified our single cell RNA-seq data into luminal A, luminal B, HER2-enriched, and basal-like subtypes based on their transcriptome profiles using the established PAM50 method⁶ (**Response Fig. 12b or New Supplementary Fig. 12i**). The proportion of HER2-enriched and basal-like cells was quite low, which is consistent with the identified signature genes (**New Supplementary Fig. 12c**). Nevertheless, we observed a significantly higher proportion of luminal B subtype in cluster 5. The prognosis of breast cancer patients with luminal B subtype is worse than those with luminal A subtype⁷. This result suggests that MCM2-2A mutant cancer cells are more likely to evolve into a worse subtype.

We further analyzed the correlation between *in vitro* CUT&Tag data and *in vivo* scRNA-seq data. We screened out the promoters with downregulated H3K27me3 or dysregulated H3K4me3 in MCM2-2A mutant MCF-7 cells, whose target genes are related to tumor growth, proliferation and metastasis, and explored their impact on gene expression *in vivo* (**Response Fig. 12c or New Fig. 5f**). We found that 35% of the genes derepressed by H3K27me3 loss continued to be upregulated at 4 weeks post-transplantation. Additionally, most genes with dysregulated H3K4me3 maintained their changed expression patterns at 4 weeks post-transplantation. Furthermore, we assessed the correlation between the expression of the upregulated genes in tumors derepressed by H3K27me3 loss and survival in breast cancer patients without endocrine- or chemo-treatment. We found that the upregulation of a proportion of these derepressed genes was associated with poor prognosis in patients (**Response Fig. 12d or New Supplementary Fig. 13c**). Moreover, a proportion of dysregulated genes in tumors targeted by dysregulated H3K4me3 were also related to poor prognosis (**Response Fig. 12e or New Supplementary Fig. 13d**). These results suggest that the dysregulation of genes targeted by the impaired H3K27me3 or H3K4me3 distribution is associated with poor prognosis in breast cancer patients.

Response Figure 12

Response Figure 12. Impaired histone inheritance leads to dysregulated gene expression correlated with poor prognosis.

a Heatmap showing the average function state score of cells in each cluster. The information of function state of breast cancer single cells was obtained from CancerSEA⁵ (<http://biocc.hrbmu.edu.cn/CancerSEA/>). **b** Proportion of single cells assigned to each breast cancer

subtype. **c** Heatmap showing the changes of H3K27me3 (*left*) and H3K4me3 (*right*) signal at promoters and the expression changes of their target genes post-transplantation. Growth-, proliferation- or metastasis-related genes with H3K27me3 downregulated or H3K4me3 dysregulated promoters were displayed. The fold change of RNA expression is calculated by comparing the average expression of each single cell in MCM2-2A mutant tumor to WT tumor collected 4 weeks post-transplantation. **d, e** Kaplan–Meier survival analysis of breast cancer patients without endocrine- or chemo-treatment stratified by the expression of selected genes. The dysregulated genes in tumors derepressed by H3K27me3 loss (**d**) or targeted by dysregulated H3K4me3 (**e**) are associated with poor prognosis in breast cancer patients. The correlation between gene expression and survival in breast cancer patients was analysed by the Kaplan-Meier plotter online (<http://kmplot.com/analysis/>).

Reviewer #2 (Remarks to the Author):

In this article, the authors constructed a MCM2-2A mutation in MCF-7 breast cancer cells and examined the impaired histone inheritance. They found the repressive histone mark H3K27me3 was notably reprogrammed, and the correlated gene expression was altered. Finally, they transplanted the mutant cells to mice and observed abnormal tumor growth and metastasis. However, based on the presented data, it is not rigorous to directly make a conclusion that the impaired histones inheritance promotes tumor growth and metastasis

Response: We appreciate the reviewer for raising this point. We constructed MCM2-2A mutant and POLE3 KO MCF-7 cells and MCM2-90A T47D cells, all of which showed impaired histone inheritance without disturbing the DNA replication function. We observed significant epigenetic reprogramming, including the repressive histone mark H3K27me3 and the active histone mark H3K4me3. The target genes of altered H3K27me3 and H3K4me3 were involved in the pathways related to cell growth, epithelial cell proliferation and EMT, which are implicated to promote tumor growth and metastasis. And we observed significantly accelerated growth in MCM2-2A mutant and POLE3 KO MCF-7 organoids, as well as in MCM2-90A T47D organoids (**Response Fig. 13 or New Supplementary Fig. 11a, b**), which is consistent with the phenotype of MCM2-2A mutant tumors *in vivo*.

We further analyzed the correlation between *in vitro* CUT&Tag data and *in vivo* scRNA-seq data. We screened out the promoters with downregulated H3K27me3 or dysregulated H3K4me3 in MCM2-2A mutant MCF-7 cells, whose target genes are related to tumor growth, proliferation and metastasis, and explored their impact on gene expression *in vivo* (**Response Fig. 14a or New Fig. 5f**). We found that 35% of the genes derepressed by H3K27me3 loss continued to be upregulated at 4 weeks post-transplantation. Additionally, most genes with dysregulated H3K4me3 maintained their changed expression pattern at 4 weeks post-transplantation. CUT&Tag-qPCR⁸ of H3K27me3 and RT-qPCR of tumor tissues further verified that the derepressed genes in MCM2-2A mutant MCF-7 cells continued to be upregulated in tumors post-transplantation (**Response Fig. 14b or New Fig. 5g**). Furthermore, the correlation between the dysregulated histone marks and their target proliferation-related genes in MCM2-2A mutant and POLE3 KO MCF-7 organoids further suggests the important role of impaired histone inheritance in tumor growth (**Response Fig. 15 or New Supplementary Fig. 13a, b**). All the findings indicate that impaired histone inheritance promotes tumor progression through epigenetic reprogramming.

Response Figure 13

Response Figure 13. Impaired histone inheritance promotes the tumor growth in MCF-7 and T47D organoids.

a The growth curve of the MCM2-2A mutant, POLE3 KO and WT MCF-7 organoids. **b** The growth curve of the MCM2-90A and WT T47D organoids.

Response Figure 14

Response Figure 14. Dysregulated histone marks in MCM2-2A mutant MCF-7 cells affect the expression of proliferation- or metastasis-related genes post-transplantation.

a Heatmap showing the changes of H3K27me3 (*upper*) and H3K4me3 (*lower*) signal at promoters and the expression changes of their target genes post-transplantation. Growth-, proliferation- or metastasis-related genes with H3K27me3 downregulated or H3K4me3 dysregulated promoters were displayed. The fold change of RNA expression is calculated by comparing the average expression of each single cell in MCM2-2A mutant to WT tumor collected 4 weeks post-transplantation. **b** CUT&Tag-qPCR for H3K27me3 occupancy at the promoter regions of selected genes [*blue bold* in (**a**)], and corresponding gene expression measured by RT-qPCR in MCM2-2A mutant and WT MCF-7 tumors. Error bars, SD (n = 4). * P < 0.05, ** P < 0.01; Student's t test.

Response Figure 15

Response Figure 15. The impact of dysregulated histone marks on the expression of proliferation-related genes in MCM2-2A mutant and POLE3 KO MCF-7 organoids.

a Heatmap showing the fold change of CUT&Tag signal at H3K27me3 downregulated (*upper*) or H3K4me3 dysregulated (*lower*) promoters whose target genes are related to tumor growth, proliferation or metastasis, and their target genes' expression in MCM2-2A and WT MCF-7 organoids. **b** Heatmap showing the fold change (POLE3 KO/WT) of H3K27me3 signal at H3K27me3 downregulated promoters whose target genes are related to tumor growth, proliferation or metastasis, and their target genes' expression in POLE3 KO and WT MCF-7 organoids.

1. To verify the accuracy of data that obtained from RNA-seq and ChIP-seq, RT-qPCR and ChIP-qPCR should be considered to perform. Especially for the selected overlapped genes.

Response: We thank the reviewer for pointing this out. We selected a number of H3K27me3 dysregulated regions whose target genes were differentially expressed and related to the tumor growth or metastasis for verification. We performed the CUT&Tag-qPCR⁸ to detect the H3K27me3 occupancy changes at representative stable, downregulated and upregulated H3K27me3 regions previously identified by CUT&Tag sequencing analysis (**Response Fig. 16a, c or New Fig. 2f, h**). The expression of their target genes was also assessed by RT-qPCR to verify the RNA-seq data (**Response Fig. 16b, c or New Fig. 2g, h**). These qPCR results matched well with our CUT&Tag and RNA-seq data.

Response Figure 16

Response Figure 16. Expression of genes targeted by dysregulated H3K27me3.

a CUT&Tag-qPCR for H3K27me3 occupancy at selected regions in MCM2-2A mutant and WT MCF-7 cells (n = 4). **b** mRNA levels measured by RT-qPCR in MCM2-2A mutant and WT MCF-7 cells. Error bars, SD (WT, n = 3, MCM2-2A, n = 4). * $P < 0.05$, ** $P < 0.01$; Student's t test. **c** Heatmap showing the fold change of H3K27me3 signal and RNA expression identified by CUT&Tag sequencing and RNA-seq, respectively.

2. To prove the impaired parental histone inheritance in MCM2-2A cell lead to abnormal tumor growth and metastasis, or clone formation, related cellular functional experiments should be considered to carry out. Also, involved gene expression should be confirmed by RT-qPCR and WB.

Response: We performed several cellular functional experiments under 2D condition and found that MCM2-2A mutant MCF-7 cells did not display accelerated proliferation (clone formation assay, EdU proliferation assay or CCK8 proliferation assay) or metastasis (scratch wound-healing migration assay) phenotype (**Response Fig. 17**). However, under 3D condition, the MCM2 mutant MCF-7 and T47D organoids grew significantly faster than WT (**Response Fig. 18a, b** or **New Supplementary Fig. 11a, b**). Similarly, POLE3 KO MCF-7 organoids also acquired growth advantages compared to WT (**Response Fig. 18a** or **New Supplementary Fig. 11a**). The percentage of proliferating cells was higher in both MCM2-2A mutant and POLE3 KO MCF-7 organoids than WT (**Response Fig. 18c, d** or **New Supplementary Fig. 11c, d**). We also observed similar changes in gene expression between POLE3 KO and MCM2-2A mutant MCF-7 organoids (**Response Fig. 18e-g** or **New Supplementary Fig. 11e-g**), and the changed genes were mainly involved in cell growth, proliferation, migration, mammary gland development and apoptotic process (**Response Fig. 18f** or **New Supplementary Fig. 11f**), which is consistent with the accelerated growth phenotypes. We also validated the expression changes of genes regulating cell proliferation by RT-

qPCR (Response Fig. 18g or New Supplementary Fig. 11g).

We tried to explain the different phenotypes of MCM2-2A mutant cells *in vitro* and *in vivo*. Impaired histone inheritance resulted in epigenetic variability, which plays a vital role in adapting to complicated environments. Under the condition of normal 2D culture, the cells faced little challenge and did not show an obvious phenotype. However, some specific altered chromatin state may confer MCM2-2A cells adaptive advantages when they encounter environmental stimuli, such as hypoxia and hormone stimuli. That is the possible reason why MCM2-2A cells accelerated growth and metastasis only when they were transplanted into the mammary gland. Even a mimic *in vivo* growth environment provided by organoid culture could confer tumor cells with impaired histone inheritance more growth advantages. However, the detailed mechanism needs further investigation, thus, we did not show the results of 2D culture in our manuscript.

Response Figure 17

Response Figure 17. The proliferation and migration of MCM2-2A mutant MCF-7 cells under the condition of 2D culture.

a Colony-formation assay of MCM2-2A mutant and WT MCF-7 cells. **b** Cell proliferation of MCM2-2A mutant and WT MCF-7 cells, evaluated by EdU staining. Error bars, SD (WT, n = 14; MCM2-2A, n = 12). ns $P > 0.05$. Mann-Whitney test. **c** The growth curve of MCM2-2A mutant and WT MCF-7 cells, detected by Enhanced Cell Counting Kit-8. **d** Scratch wound healing assay of MCM2-2A mutant and WT MCF-7 cells. Wound healing percentage = (0h Area - 48h Area) / 0h Area. Error bars, SD (n = 11). ns, $P > 0.05$. Student's t test.

Response Figure 18

Response Figure 18. Impaired histone inheritance promotes the tumor growth in MCF-7 and T47D organoids.

a The growth curve of the MCM2-2A mutant, POLE3 KO and WT MCF-7 organoids. Error bars, SD (n = 4). *** $P < 0.001$, * $P < 0.05$. One-way ANOVA test. **b** The growth curve of the MCM2-90A and WT T47D organoids. Error bars, SD (n = 4). * $P < 0.05$, Student's t test. **c** Images showing the morphology and cell alive state of MCM2-2A mutant, POLE3 KO and WT MCF-7 organoids. Propidium iodide, PI (+) represents dead cells, Hoechst (+) represents alive cells. **d** Cell proliferation evaluated by EdU staining in MCM2-2A mutant, POLE3 KO and WT MCF-7 organoids. **e** Heatmap showing the differentially expressed genes in MCM2-2A mutant vs. WT MCF-7 organoids and their corresponding expression in POLE3 KO and WT MCF-7 organoids. #1 and #2 indicate two independent clones. R1 and R2 indicate two replicates. **f** GO enrichment for the dysregulated genes in both MCM2-2A mutant and POLE3 KO MCF-7 organoids. **g** RT-qPCR showing the expression of proliferation-related genes in MCM2-2A mutant, POLE3 KO and WT MCF-7 organoids. The

genes' upregulation (*left*) or downregulation (*right*) could promote cancer cell proliferation. Error bars, SD (WT and POLE3 KO, n = 4; MCM2-2A, n = 8). * $P < 0.05$, ** $P < 0.01$, *** $P < 0.001$; Student's t test.

3. The authors did not describe the statistic method of Kaplan-Meier survival analysis, while the curves were crossed over, and the log-rank test may not be appropriate (Figure 4e).

Response: We appreciate the reviewer for this suggestion. We did use log-rank test in the previous Kaplan-Meier survival analysis. We obtained new data during the revision (6 mice in each group). Then, we included these data with our previous data and re-analyzed the survival curves (**Response Fig. 19 or New Fig. 4e**). Both log-rank test ($P = 0.016$) and Tarone-Ware test ($P = 0.047$) indicate a significant difference between two groups. Since the curves do not cross over anymore, we keep the log-rank test for the survival curve.

Response Figure 19. Kaplan–Meier survival analysis of mice bearing MCM2-2A mutant and WT tumors (WT, n = 15; MCM2-2A, n = 16). $P = 0.016$, log-rank test.

Minor errors:

The grammars and tenses should be carefully checked.

Response: We appreciate the reviewer for the criticism. We have checked and revised the grammars and tenses carefully.

Eg: Line 154/178, they should be past tense.

Response: We appreciate the reviewer for the criticism. We have revised the sentences in our manuscript. **Line 141.** “Previous studies indicated that propagations of repressive marks, such as H3K27me3 and H3K9me3 rely on a “read and write” mechanism.” **Line 175.** “Considering that upregulated H3K27me3 peaks were mainly distributed in the closed chromatin context.”

Eg: Line 142, redundant “both”.

Response: We appreciate the reviewer for the criticism. We have revised the sentence in our manuscript. **Line 128-129.** “It has been found that MCM2 facilitates the recycling of parental histones to lagging strands, whereas POLE3 promotes the recycling to leading strands in both *Saccharomyces cerevisiae* and mESCs.”

Eg: Line 197-198, the description of “The expression level of genes with H3K4me3 stable promoters was also higher in both MCM2-2A mutant and WT cells” is not accurate.

Response: We appreciate the reviewer for the criticism. We have revised the sentence in our manuscript. **Line 196-197.** “The expression level of genes with H3K4me3 stable promoters was also higher than those with H3K4me3 up- or downregulated promoters in these three models.”

The part of introduction should be described in a more concise way.

Response: We appreciate the reviewer for the criticism. We have removed the redundant sentences and revised our introduction to make it more concise. **Line 53-84.**

Reviewer #3 (Remarks to the Author):

In this paper, Tian et al describes how impairment of histone inheritance influences gene expression and tumor progression using MCF-7 breast cancer cell line. Authors used MCM2 mutant cell line in which MCM gene's histone chaperone function is specifically disturbed (without disturbing helicase function) and analyzed gene expression changes in detail and its effect on clonal evolution. Given the fact that the gene MCM2 is involved in cell proliferation in various cancers, this study provides an important information as to how MCM2 plays a role in cancer progression. This study provides several other interesting points regarding the hierarchy of gene regulation among different histone modifications and other chromatin states (open/closed, A/B compartments). The authors used new and suitable techniques to analyze genome-wide changes of these states.

Overall, the findings should be of great interest to the readers of Nature Communications. There are some concerns about data interpretation and presentation and lack of some details of analysis. I hope my comments help the authors to make this paper more compelling.

We thank the reviewer's very positive comments for our manuscript.

Major points;

1. I suggest that each result could be provided more thoroughly without fitting to certain idea first, then give interpretations later in discussion. It is especially interesting that authors observed that majority of H3K27me3 peaks remained stable in MCM2-2A mutant cells throughout multiple cell divisions, and downregulation of H3K27me3 mainly occurs at the A-compartment. From this point of view, I have an impression that histone inheritance may not be a major factor for maintaining gene expression memory. Further interpretation and discussion will be valuable for this point.

Response: We appreciate the reviewer for this suggestion. We have reorganized the writing order and further discussed the additional factors participating in gene expression memory maintenance in the revised manuscript. **Line 340-361.** "In the MCM2-2A mutant MCF-7 cells, although the transfer of parental histones to the lagging strand was blocked during replication, more than half of H3K27me3 remained stable (Supplementary Table 2). This result suggests that the restoration of H3K27me3 in MCF-7 cells is not solely dependent on the inheritance of parental histones and read-and-write propagation. In recent years, it has been discovered that DNA sequence, DNA methylation and chromatin high-order structure could also contribute to H3K27me3 restoration⁹. The accurate recycling of H2AK119ub1 during DNA replication, which is not disturbed in MCM2-2A mutant cells, could facilitate PRC2 recruitment and guide the accurate restoration of H3K27me3¹⁰. PRC2 could re-establish H3K27me3 pattern de novo with high fidelity genome-wide in mESCs¹¹, but the recovery of H3K27me3 patterns is incomplete in the neural progenitor cells (NPCs) and immortalized mouse embryonic fibroblasts (iMEFs)¹². Therefore, the inheritance of parental histones carrying H3K27me3 is necessary for maintaining H3K27me3 at specific regions in differentiated cells. Besides, our study indicates that transcription plays a vital role in reprogramming H3K27me3 landscape in MCM2-2A mutant cancer cells. Transcriptional memory hinges on the double-negative feedback between Polycomb-mediated silencing and active transcription, and its activation is a widespread developmental feature of PRC2 target genes that

emerges during cell differentiation¹². Loss of H3K27me3 in MCM2-2A mutant cancer cells appears to allow the nearby transcriptionally active state to invade formerly repressive genomic loci and activate the correspondent gene expression. Subsequently, active transcription at these aberrantly activated loci would antagonize Polycomb activity and be sustained in a transcription-dependent way. The failure of H3K27me3 restoration across cell divisions can re-activate mammary gland development processes (Fig. 2d) that are often hijacked by breast cancer cells as drivers for tumor progression^{13, 14}.”

We have included this discussion in the revised manuscript. **Line 380-385**. “Although the influence of impaired histone inheritance on the transcriptionally active regions is minor than repressive regions, we observed the persisting effects of dysregulated H3K4me3 on gene expression post-transplantation, suggesting an important role of H3K4me3 reprogramming in tumor progression. We focused on H3K27me3 reprogramming in this article, while the dramatic changes of enhancers and H3K9me3 marked heterochromatin regions could also participate in driving tumor progression.”

2. MCM-2 mutant cells seem to have “lost” template H3K27me3 from actively transcribed regions, and show quite fixed/stable phenotype. These cells do not seem to “keep forgetting” of their memory further. In that case, what explain the mechanism of tumor evolution? MCM-2 mutant may have more phenotypic (gene expression level) or epigenetic variability than WT MCF7 cells? I believe that single cell data can be analyzed to consider the mechanism deeper.

Response: We appreciate the reviewer for the thoughtful suggestions.

The H3K27me3 and gene expression patterns reflect both genetic and epigenetic information. Although the MCM2-2A mutation affects the epigenetic inheritance process, especially for actively transcribed regions, it does not fully lose epigenetic inheritability. As only the transferring of parental H3-H4 tetramers to the lagging strands is disrupted, parts of the H3K27me3 pattern memory maintained through DNA sequence-mediated PRC2 targeting, H2AK119ub1 recycling¹⁰ and other epigenetic feathers. However, due to the loss of tight epigenetic inheritance control, MCM2-2A MCF-7 cells may obtain more epigenetic variability. It is possible that some mutant cells obtain aberrant H3K27me3 profiles favoring adaptation to the *in vivo* environment and then drive tumor evolution. For instance, we found that a proportion of proliferate- or metastasis- related genes derepressed by H3K27me3 loss kept upregulated at 4 weeks post-transplantation (**Response Fig. 20 or New Fig. 5f, g**). The correlation between the aberrant distribution of histone marks *in vitro* and the dysregulation of their target genes *in vivo* implied the important role of epigenetic variability in tumor evolution.

On the other hand, to investigate the role of transcriptional variability on MCM2-2A mutant tumor evolution, we further analyzed *in vitro* scRNA-seq data of WT and MCM2-2A mutant MCF-7 cells. Based on their transcriptome profiles, we resolved the cells into five clusters with two clusters in WT cells and three clusters in MCM2-2A mutant cells (**Response Fig. 21a or New Fig. 6c**), suggesting higher transcriptional diversity of MCM2-2A mutant cells. Notably, cluster 2 cells expressed high level of genes regulating DNA replication, mitotic cell cycle and cell division, indicating growth advantages (**Response Fig. 21b or New Fig. 6d**). Of note, a few proliferation-promoting genes derepressed by H3K27me3 loss were highly expressed in cluster 2, such as POP7¹⁵, TNNT1¹⁶ and TYMS¹⁷. Moreover, cluster 2 cells also expressed high levels of genes involved in chromatin remodeling and oxidative phosphorylation (**Response Fig. 21b or New Fig. 6d**). It has been shown that increased oxidative phosphorylation is critical for lung metastasis in the human

breast cancer¹⁸. In addition, cells in cluster 4 expressed high levels of genes regulating epithelial cell migration and differentiation (**Response Fig. 21c or New Supplementary Fig. 14g**), which could potentially also promote lung metastasis. We then reconstructed the lineage relationships between the cells *in vitro* and the tumor cells collected 4 weeks post-transplantation based on their LARRY barcodes (**Response Fig. 21d or New Fig. 6e**). We annotated cells that would grow into dominant clones in MCM2-2A tumors and found that they mainly originated from cluster 2 *in vitro* (**Response Fig. 21e or New Fig. 6f**). Given the function of highly expressed genes in cluster 2 *in vitro*, we speculate that some MCM2-2A mutant cells might have already obtained the ability of accelerated proliferation *in vitro*.

Based on these results, we propose that the epigenetic variability and gene expression variation complement each other and corporately drive MCM2 mutant tumor evolution.

Response Figure 20

Response Figure 20. Dysregulated histone marks in MCM2-2A mutant MCF-7 cells effects the expression of proliferation- or metastasis-related genes post-transplantation.

a Heatmap showing the changes of H3K27me3 (upper) and H3K4me3 (lower) signal at promoters and the expression changes of their target genes post-transplantation. Growth-, proliferation- or metastasis-related genes with H3K27me3 downregulated or H3K4me3 dysregulated promoters were displayed. The fold change of RNA expression is calculated by comparing the average expression of each single cell in MCM2-2A mutant tumor to WT tumor collected 4 weeks post-transplantation. **b** CUT&Tag-qPCR for H3K27me3 occupancy at the promoter regions of selected genes [blue bold in (a)], and corresponding gene expression measured by RT-qPCR in MCM2-2A mutant and WT MCF-7 tumors. Error bars, SD (n = 4). * P < 0.05, ** P < 0.01; Student's t test.

Response Figure 21

Response Figure 21. Colony evolution in MCM2-2A mutant cells.

a scRNA-seq of MCM2-2A mutant and WT MCF-7 cells *in vitro*. Cells were dissolved into 5 clusters. Clusters 2, 3 and 4 were mainly composed of MCM2-2A cells, and clusters 1 and 5 were composed of WT cells. **b** Gene ontology enrichment analysis for marker genes of cluster 2 *in vitro*. Green arrows represent cell proliferation and blue arrow represents metabolism reprogramming. **c** Gene ontology enrichment analysis for marker genes of cluster 4 *in vitro*. **d** Barcode abundance (clone size) in the tumor cell population prior to engraftment (*in vitro*) and in tumors 4 weeks post-transplantation (*in vivo*). Barcodes are ordered vertically according to their initial abundance (highest to lowest from top to bottom) *in vitro*, only the barcodes that are simultaneously captured both *in vitro* and *in vivo* are shown. The number of clones that would grow into dominant clones from MCM2-2A MCF-7 cells is far more than that from WT cells. **e** Contour plots projected onto *t*-SNE plot showing cell density of *in vitro* MCM2-2A mutant cells that would grow into dominant clones.

3. Can authors provide more logic of each analysis and how to read graphs more for general audiences. For example, how eSPAN graph is made, what the “bias” is, would be helpful if they can spell out more clearly in the text or figure legends.

Response: We appreciate the reviewer for this suggestion. We have added a more detailed figure

legends explaining how to read our analysis results in New Fig. 1b, New Fig. 6e, New Supplementary Fig. 4a and New Supplementary Fig. 12g. Besides, we added a graph explaining the analysis of the eSPAN procedure in New Supplementary Fig. 1d or Response Fig. 22 and interpreted how to read the results in details in the figure legends.

Response Figure 22

Response Figure 22. H3K36me3 eSPAN analysis.

We analyzed the impact of MCM2-2A mutation on nucleosome assembly using surrogate marks of parental histones (H3K36me3) as reported previously.^{19, 20} The bias of H3K36me3 eSPAN at selected replication origins ($n = 7,624$) was computed from unique fragments in each bin using the formula $Bias = (W - C) / (W + C)$; where W and C are the number of fragments mapped onto the Watson and Crick strands in each bin, respectively. There is no bias or a slight leading bias in WT cells, representing the parental histone H3K36me3 are symmetrically recycled to the leading strands and the lagging strands. There is a leading bias in MCM2-2A mutant cells, which means parental histones recycled to the leading strands are more than that to the lagging strands.

4. It is a good design to use MCM2 tyrosine mutants to preserve helicase function. It will be informative if they can show MCM2 mutation in homozygous for their lines (sanger sequence data) and show helicase function (cell division?) is not perturbed in those obtained clones.

Response: We appreciate the reviewer for their thoughtful suggestions. We have showed the Sanger sequence data of MCM2 homozygous mutation (Response Fig. 23a or New Supplementary Fig. 1a). In the revised manuscript, we have investigated the DNA replication function of the mutant MCM2 by EdU staining. We observed similar EdU incorporation between WT and MCM2-2A

mutant MCF-7 cells (**Response Fig. 23b or New Supplementary Fig. 1c**). This data indicates that the helicase and DNA replication function of the mutant MCM2 is not disturbed in MCM2-2A mutant MCF-7 cells. This is consistent with previous studies in mESCs (Petryk *et al.* Science. 2018) and budding yeast (Foltman *et al.* Cell Rep. 2013). Foltman *et al.* showed that DNA synthesis was normal in *mcm2-2A* or *mcm2-3A* mutant budding yeast strains.²¹ Petryk *et al.* also showed that disrupted histone-binding function of MCM2 (*MCM2-2A*) did not affect cell cycle progression in mice ESCs.²²

Response Figure 23

Response Figure 23. Helicase and DNA replication function is not perturbed in MCM2-2A mutant MCF-7 cells.

a Sanger sequencing confirms homozygous MCM2-2A mutation of selected colony. **b** EdU staining of MCM2-2A and WT MCF-7 cells. Cells were treated with EdU at 10 μ M for 2 h. The number of EdU-positive cells and all the cells (marked by DAPI) was determined with ImageJ software. Error bars, SD (WT, n = 14; MCM2-2A, n = 12). ns $P > 0.05$. Mann-Whitney test.

5. It should be informative if authors can provide any data or information why they used H3K36me3 to detect preexisting histones in their cells if possible.

Response: We thank the reviewer for raising these points. As previously reported, post-translational modification H3K36me3 marks parental histones and is faithfully inherited to daughter strands during DNA replication²³. Since it takes nearly a whole cell cycle for newly deposited histone H3 to restore the H3K36me3 abundance²⁴, H3K36me3 was commonly used in eSPAN experiments to monitor the parental histones H3 at leading and lagging strands of DNA replication forks^{19, 20}. And we have added the statement to our manuscript (**Line 96-98**).

6. Based on the method section, eSPAN data was normalized by BrdU signal. Does the “signal” mean intensity or BrdU-IP ssDNA sequencing data? Either way, they need to provide data and details.

Response: We thank the reviewer for pointing out this omission. BrdU signal means the intensity of BrdU-IP ssDNA sequencing data. Because the potential asymmetric incorporation of BrdU at leading and lagging strands arising from uneven distribution of nucleotides at specific genomic regions, we used the BrdU-IP ssDNA-seq for normalization. We have supplied the experiment details in Methods (**Line 477-509**) and uploaded the BrdU-IP ssDNA sequencing data to the GEO database (GSE201262).

7. It would be informative how many origins out of selected replication origins are active (or inactive) in their lines based on their brdu seq data and if so, is there any correlation between gene activity change relative to the distance from the origin?

Response: We appreciate the reviewer for this suggestion. However, our BrdU-ip seq data is not suitable for screening out the active and inactive origins. Because the cells for BrdU-IP ssDNA sequencing were taken from the cells for eSPAN experiment. To obtain enough eSPAN products, the time for BrdU treatment need to be as long as 1 hour in nonsynchronized MCF-7 cells. The BrdU labeled genomic DNA was fragmented using pA-Tn5-ME-A adaptor complex, and then the newly synthesized DNA strands were enriched by BrdU-IP. Therefore, our BrdU-IP data is far different from the studies investigating replication origins, which separated the BrdU-labeled Okazaki fragments from parental strands by boiling for 3 to 10 min.^{25, 26} Thus, it is not possible for us to distinguish the active and inactive origins using our BrdU seq data.

We analyzed the distance from the TSS of each gene to its nearest origin (previously published data²⁶, n = 7,624), and found no obvious difference in the distance between the differentially expressed genes and stable expressed genes. The activity change of a gene was not relative to its distance from the origin (**Response Fig. 24**), but there may be some deviations between the origin locations we used and the actual active replication origins. Further well-designed investigations are needed to verify this conclusion.

Response Figure 24

Response Figure 24. Distance from the TSS of each gene to its nearest origin. Error bars, SD. Student's t test.

8. MCM2 mutant lines show increased H3K27me3 signal in western blotting. If that is the case, did they normalize the H3K27me3 cut and tag data with the protein amount?

Response: We thank the reviewer for raising this point. In our manuscript, we did not normalize the CUT&Tag data with protein amount, because western blot is a semi-quantitative experiment. We normalized the CUT&Tag data to library size (Fragments Per Kilobase per Million mapped fragments, FPKM). To verify the reliability of our data analysis, we performed the CUT&Tag-qPCR to detect the H3K27me3 occupancy changes at representative stable, downregulated and upregulated H3K27me3 regions previously identified by CUT&Tag sequencing analysis. The

results of qPCR matched well with our CUT&Tag sequencing data (**Response Fig. 25a, b**). Besides, we used the small amount of tracer genomic DNA derived from the *E. coli* during transposase protein production to normalize sample read counts²⁷ and identified bins that differed between MCM2-2A mutant and WT cells. Then we evaluated the correlation of dysregulated bins identified by these two normalization methods (**Response Fig. 25c or New Supplementary Fig. 2a**). The high correlation ($R^2 = 0.8987$; $P < 0.0001$; Pearson correlation test) suggests that the results from the two normalization methods are comparable. Taken together, normalized to library size is suitable for our H3K27me3 CUT&Tag sequencing data.

Response Figure 25

Response Figure 25. Verification of the H3K27me3 dysregulated peaks.

a Heatmap showing the fold change of H3K27me3 signal detected by CUT&Tag and normalized to library size. **b** CUT&Tag-qPCR for H3K27me3 occupancy at selected regions in MCM2-2A mutant and WT MCF-7 cells. Error bars, SD (n = 4). * $P < 0.05$, ** $P < 0.01$, *** $P < 0.001$; Student's t test. **c** Dotplot showing the correlation of two normalization methods. *x*-axis shows the fold change of H3K27me3 in each bin following the normalization to library size (FPKM), *y*-axis shows the fold change of H3K27me3 in each bin following the normalization to the spike-in genomic DNA derived from the *E. coli* during transposase protein production.

9. Their definition of “downregulated” “upregulated” and “stable” loci is confusing in the texts. Please give the definition more clearly for each place.

Response: We appreciate the reviewer for this reminding. We have checked the statement throughout the manuscript and modified the following places:

Line 117-119. “Downregulated H3K27me3 peaks were significantly enriched at promoters, 5’ UTRs, exons, transcription termination sites (TTSs), and CpG island regions in MCM2-2A mutant cells compared to the WT cells”.

Line 151-153. “We compared the original chromatin state at altered H3K27me3 regions in MCF-7 cells, and observed significant H3K36me3 enrichments flanking the H3K27me3 downregulated peaks resulting from MCM2-2A mutation”.

Line 172-173. “As expected, the expression level of genes with downregulated H3K27me3 at promoters increased significantly”.

Line 190-191. “We investigated the original epigenetic profiles of the upregulated, stable, and downregulated H3K4me3 peaks in three MCM2 mutant cells”.

Besides, we have defined the “downregulated”, “upregulated” and “stable” regions in our Methods

Line 589-591. “Upregulated” means “ P -value < 0.01 and Log_2 [fold change (MCM2-2A/WT)] > 1 ”; “downregulated” means “ P -value < 0.01 and Log_2 [fold change (MCM2-2A/WT)] < -1 ”; “stable” means any other regions with indicated histone marks.

References

1. Semina SE, *et al.* Identification of a novel ER-NFkB-driven stem-like cell population associated with relapse of ER+ breast tumors. *Breast Cancer Res* **24**, 88 (2022).
2. Hong SP, *et al.* Single-cell transcriptomics reveals multi-step adaptations to endocrine therapy. *Nat Commun* **10**, 3840 (2019).
3. Semina SE, *et al.* Selective pressure of endocrine therapy activates the integrated stress response through NFkappaB signaling in a subpopulation of ER positive breast cancer cells. *Breast Cancer Res* **24**, 19 (2022).
4. Chen F, *et al.* Single-Cell Transcriptomic Heterogeneity in Invasive Ductal and Lobular Breast Cancer Cells. *Cancer Res* **81**, 268-281 (2021).
5. Yuan H, *et al.* CancerSEA: a cancer single-cell state atlas. *Nucleic Acids Res* **47**, D900-D908 (2019).
6. Parker JS, *et al.* Supervised risk predictor of breast cancer based on intrinsic subtypes. *J Clin Oncol* **27**, 1160-1167 (2009).
7. Pu M, *et al.* Research-based PAM50 signature and long-term breast cancer survival. *Breast Cancer Res Treat* **179**, 197-206 (2020).
8. Du G, *et al.* The accessible promoter-mediated supplementary effect of host factors provides new insight into the tropism of SARS-CoV-2. *Mol Ther Nucleic Acids* **28**, 249-258 (2022).
9. Yang Y, Li G. Post-translational modifications of PRC2: signals directing its activity.

Epigenetics Chromatin **13**, 47 (2020).

10. Flury V, *et al.* Recycling of modified H2A-H2B provides short-term memory of chromatin states. *Cell* **186**, 1050-1065 e1019 (2023).
11. Hojfeldt JW, *et al.* Accurate H3K27 methylation can be established de novo by SUZ12-directed PRC2. *Nat Struct Mol Biol* **25**, 225-232 (2018).
12. Holoch D, *et al.* A cis-acting mechanism mediates transcriptional memory at Polycomb target genes in mammals. *Nat Genet* **53**, 1686-1697 (2021).
13. Holliday H, Baker LA, Junankar SR, Clark SJ, Swarbrick A. Epigenomics of mammary gland development. *Breast Cancer Res* **20**, 100 (2018).
14. Vafaizadeh V, Peuhu E, Van Keymeulen A, Koledova Z. Editorial: Perspectives in Mammary Gland Development and Breast Cancer Research. *Front Cell Dev Biol* **8**, 719 (2020).
15. Huang Y, *et al.* RNA binding protein POP7 regulates ILF3 mRNA stability and expression to promote breast cancer progression. *Cancer Sci* **113**, 3801-3813 (2022).
16. Shi Y, *et al.* TNNT1 facilitates proliferation of breast cancer cells by promoting G(1)/S phase transition. *Life Sci* **208**, 161-166 (2018).
17. Xu W, Jiang H, Zhang F, Gao J, Hou J. MicroRNA-330 inhibited cell proliferation and enhanced chemosensitivity to 5-fluorouracil in colorectal cancer by directly targeting thymidylate synthase. *Oncol Lett* **13**, 3387-3394 (2017).
18. Davis RT, *et al.* Transcriptional diversity and bioenergetic shift in human breast cancer metastasis revealed by single-cell RNA sequencing. *Nat Cell Biol* **22**, 310-320 (2020).
19. Li Z, *et al.* DNA polymerase alpha interacts with H3-H4 and facilitates the transfer of parental histones to lagging strands. *Sci Adv* **6**, eabb5820 (2020).
20. Xu X, Duan S, Hua X, Li Z, He R, Zhang Z. Stable inheritance of H3.3-containing nucleosomes during mitotic cell divisions. *Nat Commun* **13**, 2514 (2022).
21. Foltman M, *et al.* Eukaryotic replisome components cooperate to process histones during chromosome replication. *Cell Rep* **3**, 892-904 (2013).
22. Petryk N, *et al.* MCM2 promotes symmetric inheritance of modified histones during DNA replication. *Science* **361**, 1389-1392 (2018).
23. Reveron-Gomez N, *et al.* Accurate Recycling of Parental Histones Reproduces the Histone

- Modification Landscape during DNA Replication. *Mol Cell* **72**, 239-249 e235 (2018).
24. Li F, *et al.* The histone mark H3K36me3 regulates human DNA mismatch repair through its interaction with MutSalpha. *Cell* **153**, 590-600 (2013).
 25. Karnani N, Taylor CM, Malhotra A, Dutta A. Genomic study of replication initiation in human chromosomes reveals the influence of transcription regulation and chromatin structure on origin selection. *Mol Biol Cell* **21**, 393-404 (2010).
 26. Petryk N, *et al.* Replication landscape of the human genome. *Nat Commun* **7**, 10208 (2016).
 27. Kaya-Okur HS, *et al.* CUT&Tag for efficient epigenomic profiling of small samples and single cells. *Nat Commun* **10**, 1930 (2019).

REVIEWERS' COMMENTS

Reviewer #1 (Remarks to the Author):

The authors have addressed all concerns and suggestions raised by either performing additional experiments or clarifying and expanding on analysis. The dataset is now robust for publication.

Reviewer #2 (Remarks to the Author):

Very impressive revision. I appreciate it.

Reviewer #3 (Remarks to the Author):

The authors have addressed my questions and comments raised for the previous submission. Therefore, I support acceptance of this work.